# Improving Morpho-Physiological Indicators, Yield, and Water Productivity of Wheat through an Optimal Combination of Mulching and Planting Patterns in Arid Farming Systems

**Salah El-Hendawy** \*, **Bazel Alsamin**, **Nabil Mohammed** and **Yahya Refay**

Department of Plant Production, College of Food and Agriculture Sciences, King Saud University,
P.O. Box 2460, Riyadh 11451, Saudi Arabia
\* Correspondence: mosalah@ksu.edu.sa; Tel.: +966-535318364

**Abstract:** Mulching practices (M), which conserve soil water and improve water productivity (WP), are receiving increasing attention worldwide However, so far, little attention has been given to investigating the effects of the integrations of mulching and planting patterns (IMPPs) on spring wheat performance under arid regions conditions. A two-year field study was conducted to compare the effects of eight IMPPs on growth parameters at 80 and 100 days after sowing (DAS), growth indicators, physiological attributes, grain yield (GY), and WP of wheat under adequate (1.00 ET) and limited (0.50 ET) irrigation conditions. The IMPPs included three planting patterns (PPs), that is, flat (F), raised-bed (RB), and ridge–furrow (RF), in combination with three M, that is, no-mulch (NM), plastic film mulch (PFM), and crop residues mulch (CRM). The results indicated that PPs mulched with PFM and CRM significantly increased growth indicators, different growth parameters, physiological attributes, GY, and WP by 6.9–39.3%, 8.2–29.2%, 5.2–24.9%, 9.9, and 11.2%, respectively, compared to non-mulched PPs. The F and RB patterns mulched with CRM were more effective in improving growth parameters at 100 DAS (2.7–13.6%), physiological attributes (0.2–20.0%), GY, and WP (9.7%) than were the F and RB patterns mulched with PFM under 1.00 ET, while the opposite was true under 0.50 ET conditions. Although the RFPFM failed to compete with other IMPPs under 1.00 ET, the values of different parameters in this PP were comparable to those in F and RB patterns mulched with PFM, and were 1.3–24.5% higher than those in F and RB patterns mulched with CRM under 0.50 ET conditions. Although the RFNM did not use mulch, the values of different parameters for this PP were significantly higher than those of F and RB patterns without mulch. Irrespective of irrigation treatments, the heatmap analysis based on different stress tolerance indices identified the different PPs mulched with PFM as the best IMPPs for the optimal performance of wheat under arid conditions, followed by PPs mulched with CRM. The different growth indicators exhibited second-order and strong relationships with GY ($R^2$ = 0.78 to 0.85) and moderate relationships with WP ($R^2$ = 0.59 to 0.79). Collectively, we concluded that using PPs mulched with CRM is the recommended practice for achieving good performance and production for wheat under adequate irrigation, whereas using PPS mulched with PFM is recommended as a viable management option for sustainable production of wheat and improving WP under limited irrigation in arid countries.

**Keywords:** crop residues mulching; leaf area duration; net assimilation rate; plastic film mulching; raised-bed mulching; ridge–furrow mulching; stress tolerance indices

## 1. Introduction

The issue of food security is one of the most critical issues of concern to governments around the world because of its great impact on the national security of any country, as well as the diversity of its human, social, economic, and political dimensions. Unfortunately, the agriculture sector in arid and semiarid countries, which is responsible for feeding approximately 40% of the world's population, is facing multiple and complex challenges

in terms of water scarcity, frequent drought, land degradation, and high climatic variability [1,2]. These trends are expected to increase, with future climate change driving increased frequencies of extreme climatic events. Moreover, groundwater, which is an important source of water for more than 40% of the area equipped for irrigation, is being depleted at a very fast rate and is becoming scarce in many arid and semiarid countries [3]. This means that the lives and livelihoods of many people in these countries are at immediate risk from acute food insecurity. Since inadequate irrigation water has a significant negative effect on the production of various crops, the balance between limited water supplies and food production has become an issue that needs more attention from governments in all arid and semiarid countries, if they are to achieve real food security. Therefore, conservation of irrigation water should be the slogan behind crop production in these countries, especially since the irrigation sector consumes more than 75% of the total available freshwater resources, as well as the fact that the water productivity (WP) of all field crops is very low in these countries [4–6]. Therefore, increasing the food production per unit of irrigation water applied, or WP, is the key strategy for ensuring water sustainability for future food security in arid and semiarid countries [6,7].

As the wheat crop is at the epicenter of global food security and is grown on more land area than any other crop, maximizing WP for this crop is crucial to the sustainability of wheat production systems and for food security in arid and semiarid countries. Therefore, there is an urgent need to develop and implement appropriate water-saving and conservation farming practices to achieve this goal under water scarcity conditions [6,8–10]. In arid and semiarid conditions, WP for wheat is very low, because approximately 40–60% of irrigation water is lost from the soil's surface through evaporation, especially during the early growth stages of the crop when the plants are still small and do not cover the entire surface of the soil [11–14]. Thus, any farming practices that have the potential to keep soil evaporation to a minimum and improve the soil's water content and the amount of water available to plants could be considered one of the most effective options for improving the production and WP of wheat under limited water supplies. Recently, mulching and planting patterns (PPs) have gained significant attention as effective farming practices in many countries around the world as means to effectively achieve this goal by conserving soil water during the entire crop-growth period and providing the best opportunity for an increasing crop yield and favorable WP [6,7,15–18].

Cutting off contact between the soil's surface and the atmospheric evaporation layer is the first step in reducing the amount of soil moisture lost by direct evaporation. Thus, mulching, which refers to covering the soil surface with organic and inorganic materials, is one of the best farming practices for achieving this purpose. Mulching not only reduces soil evaporation, but also improves soil's physicochemical and biological properties, regulates soil temperature, matches water supply and demand, increases infiltration and storage of water in the root zone, restricts soil erosion, increases nutrient availability, decreases the leaching loss of fertilizers around the root zone, reduces the root-zone salinity, suppresses weed infestation, lowers the population of pathogens, and promotes carbon dioxide ($CO_2$) retention in leaves [6,8,19–26]. As a result, all the above multifaceted benefits of mulching create favorable conditions which directly and indirectly exert positive impacts on crop growth and development, not only under a limited water supply, but also under sufficient water supply conditions. Previous studies have shown that, in general, plants grown with soil mulching showed greater plant height, tiller number, leaf area index (LAI), biomass accumulation, relative growth rate (RGR), photosynthetic rate, chlorophyll content, grain yield, and WP, as well as lower days-to-emergence than those grown without mulching [16,27–30]. Therefore, mulching has become a very important practice for the agriculture sector in arid and semiarid countries, as irrigation water resources are very limited. However, the effects of mulching practices on plant growth and production vary, and sometimes there are conflicting findings in the literature, likely due to differences in materials used, crop varieties, climatic conditions, water input levels, and field manage-

ment. Moreover, the use of plastic as mulch for wheat is still in the development stage in Saudi Arabia, which is a typical arid country.

Integrations of mulching with modifying PPs (IMPPs) could also be considered one of the most viable practices for efficient irrigation water use and sustainable crop production in arid countries. Recently, several IMPPs have been developed and adapted to achieve this goal by designing IMPPs in a way that helps to reduce soil evaporation and preserve surface runoff, which ultimately enhances and prolongs water availability in the root zone, as well as providing adequate water at the key growth stages of the crop. As mentioned in the literature, flat planting (F) fully mulched with crop residues (FCRM) or plastic film (FPFM), raised-bed-furrow planting (RB) as either raised-bed and furrow mulched with crop residues (RBCRM) or plastic film (RBPFM), ridge–furrow planting (RF) with ridge and furrow mulched with plastic film (RFPFM) or ridge mulched with plastic film and furrow mulched with crop residues (RPFM-FCR), and alternating ridges and furrows (ARF) with only the ridges mulched with plastic film (ARFPFM) are different designs for IMPPs [6,16,17,31–34]. As of yet, the RFPFM has been applied for the first time in rain-fed agricultural areas to improve rainwater harvesting and enhance the moisture retention capacity of soils. However, this is the first time that the RFPFM has been applied for wheat in irrigated agricultural areas under arid climatic conditions. In rain-fed agricultural areas, the RFPFM has been designed to collect rainfall and sown seeds in furrows utilizing a ridge covered with plastic film to reduce the amount of water lost through soil evaporation. Previous studies have reported that the RFPFM significantly improved soil water content, crop phenology, grain yield, and the WP of different field crops when compared with non-mulching flat planting (FNM) [7,16,35,36].

The impacts of IMPPs on the production and WP of wheat are not consistent. Moreover, the information available on the impacts of IMPPs on the agro-physiological characteristics of irrigated wheat under arid conditions is very scarce. Therefore, the main objectives of this study were to (1) investigate the effects of different IMPPs on growth indicators, the growth performance at different growth stages, physiological attributes, GY, and WP of spring wheat grown under both adequate irrigation and limited irrigation conditions, and (2) recognize the most efficient IMPPs for achieving good performance and maximizing GY and WP for spring wheat under both irrigation conditions using the association among the studied parameters and IMPPs and the relationship of growth indicators with GY and WP. The results of this study could be helpful for water management approaches in spring wheat production in the arid agro-ecosystem with limited water resources.

## 2. Materials and Methods

### 2.1. Experimental Site Description

Experiments were carried out during the winter season of 2019–2020 and 2020–2021 at the Agricultural Research Center of the College of Food and Agriculture Sciences, King Saud University, Saudi Arabia (24°25′ N and 46°34′ E). The average temperatures, humidity, and precipitation of the experimental field during the two growing seasons are shown in Figure 1. The soil texture of the experimental site is classified as sandy loam (i.e., sand, silt, and clay of 56.7%, 28.4%, and 14.9%, respectively). Soil of the experimental site was characterized by the following proprieties: bulk density, 1.48 g cm$^{-3}$; Walkley–Black C, 0.34%; organic matter, 0.46%; available N, 45.2 mg kg$^{-1}$; available K, 186.9 mg kg$^{-1}$; available P, 7.44 mg kg$^{-1}$; field water holding capacity, 18.9%; permanent wilting point, 7.3%; and available soil moisture, 11.6%.

### 2.2. Experimental Design and Treatments

The experiment was laid out in a split-plot design with three replicates. The irrigation treatments, which were based on evapotranspiration (ETc) replacement and determined according to Allen et al. [37] from the crop coefficient (Kc) and the daily reference evapo-transpiration (ETo) of wheat, were allocated in the main plots. The irrigation treatments included full (100% ETc) and limited (50% ETc) irrigation regimes. The different IMPPs

were assigned to the sub-plots. The IMPPs treatments, as shown in Figure 2, included three PPs, that is, flat (F), raised-bed (RB), and ridge–furrow (RF), in combination with three mulching practices, that is, no-mulch (NM), plastic film mulch (PFM), and crop residues mulch (CRM). Each subplot size was 4 m length × 2 m width with a 1 m buffer zone between two adjacent subplots. Each subplot included eight wheat rows with 0.2 m spacing, two raised beds with four wheat rows and 0.2 m space between rows for each bed, and four ridges with 0.5 m distance from the adjacent ridge (center–center) in PPs of F, RB, and RF treatments, respectively. Each bed was approximately 0.2 m high and each ridge was 0.2 m deep (Figure 2).

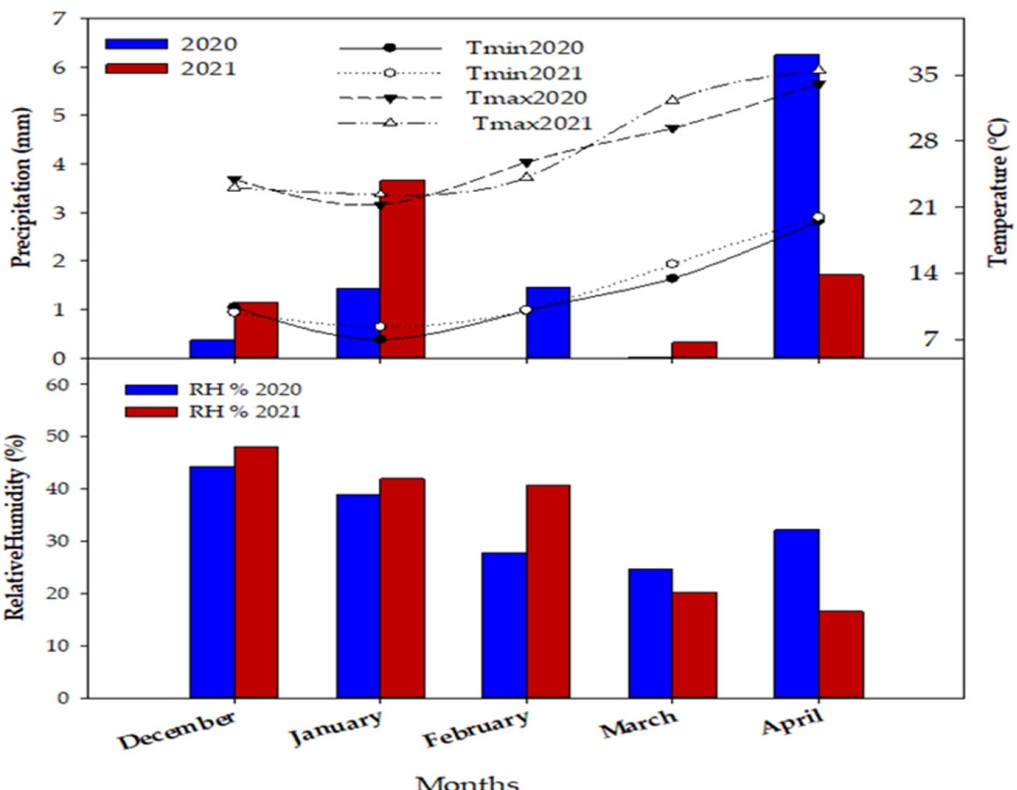

**Figure 1.** Maximum/minimum air temperature (Tmax/Tmin), precipitation, and relative humidity during the 2019–2020 and the 2020–2021 wheat growing seasons in experimental fields at the Agricultural Research Center.

The entire area of the subplot in the different-plastic-film mulched treatments was covered with plastic clear film (0.12 mm-thick) before sowing and longitudinal incisions were made in the film at 20 cm intervals to plant the seeds. In the different CRM-mulched treatments, air-dried wheat straw (5–10 cm length) was evenly distributed over the soil's surface between rows at a rate of 6000 kg ha$^{-1}$ after seedling emergence (Figure 2). The seeds of the wheat cultivar Summit were planted at a rate of 150 kg ha$^{-1}$ in all treatments. The seeds were planted on the slopes of the ridges in RF treatments, while they were buried in rows in F and RB treatments (Figure 2). Seeds were sown on 11 December and 22 November, in 2019 and 2020, respectively. Plants in all treatments were fertilized with phosphorus (calcium superphosphate, 18.5% $P_2O_5$), potassium (potassium sulfate, 48% $K_2O$), and nitrogen (ammonium nitrate, 33.5% N) at the rates of 90, 100, and 180 kg ha$^{-1}$, respectively. The entire doses of phosphorus and potassium fertilizers were applied before sowing, while nitrogen fertilizer was applied in three equal doses after sowing, and at the tillering and booting stages. The irrigation water was applied using a low-pressure surface irrigation, which consisted of a main line (76 mm in diameter) which branched off into the sub-main hoses at each subplot. Each subplot had a manual control valve to control the

amount of irrigation. The cumulative amounts of the irrigation water applied for full (100% ETc) and limited (50% ETc) irrigation regimes were 650 and 325 mm ha$^{-1}$, respectively.

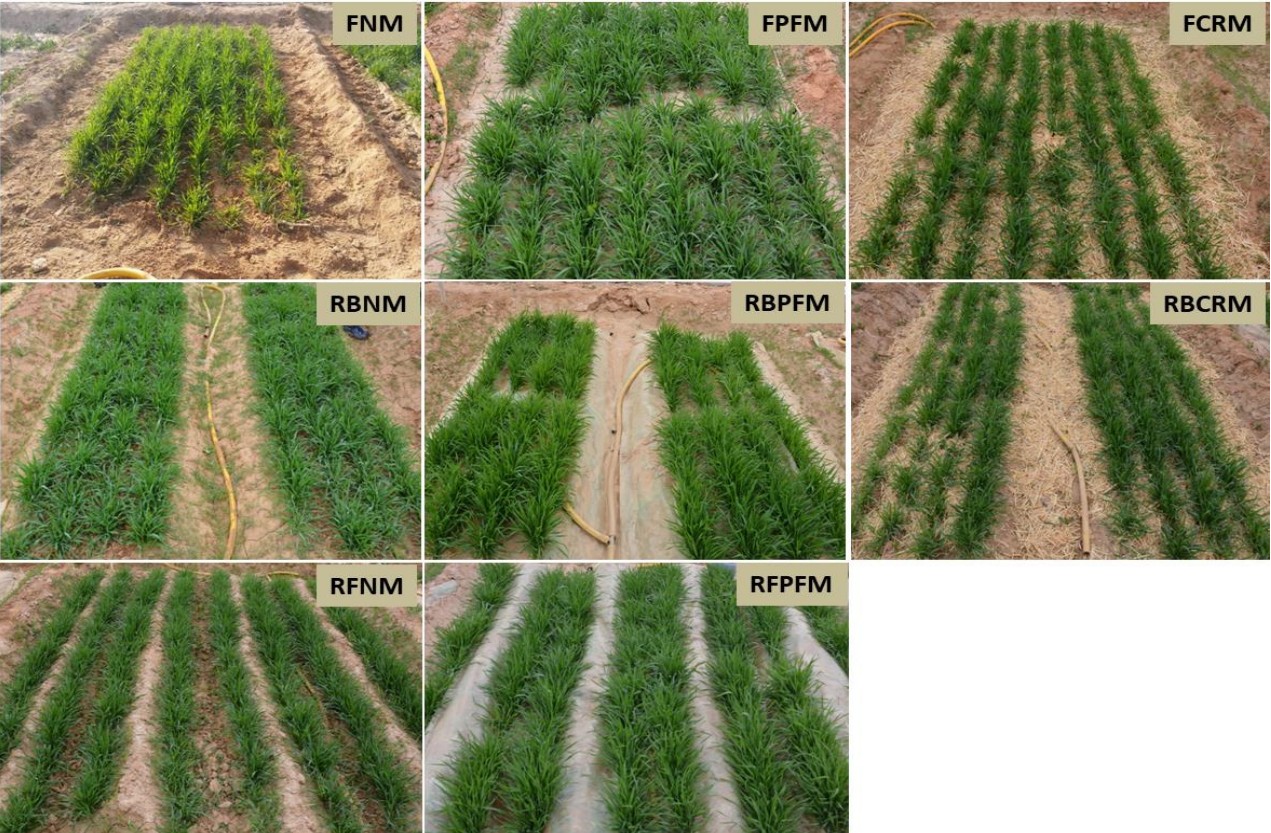

**Figure 2.** Plot layouts of different integrations of mulching and planting patterns (IMPPS). F, RB, and RF represent flat, raised-bed, and ridge–furrow planting patterns, respectively, whereas NM, PFM, and CRM represent no-mulch, plastic film mulch, and crop residues mulch, respectively.

*2.3. Measurements and Calculations*

2.3.1. Growth Parameters and Growth Indicators

At 80 and 100 days from sowing, ten plants were selected at random from each subplot to record plant height (PH), number of tillers (TN) and green leaves (GLN) per plant, green leaf area (GLA) per plant, and dry weight of leaves and total dry weight per plant (LDW and TDW). The green leaves were separated, and their blades were passed through an area meter (LI 3100; LI-COR Inc., Lincoln, NE, USA) to record the GLA. Thereafter, all leaves and stems of ten plants were oven-dried at 75 °C to a constant weight, weighed, and divided by 10 to record LDW and stem dry weight (SDW). The total dry plant biomass (PDW) per plant was obtained by the summation of LDW and SDW.

The different growth indicators, namely leaf area index (LAI), absolute growth rate (AGR), relative growth rate (RGR), net assimilation rate (NAR), leaf area duration (LAD), leaf area ratio (LAR), and crop growth rate (CGR) were calculated using the following formulas [38]:

$$\text{LAI} = \frac{\text{GLA}}{\text{Ground area}}, \tag{1}$$

$$\text{AGR}\left(\text{g day}^{-1}\right) = \frac{\text{W2} - \text{W1}}{\text{T2} - \text{T1}}, \tag{2}$$

$$\text{RGR}\left(\text{g g}^{-1}\text{ day}^{-1}\right) = \frac{\text{Ln W2} - \text{Ln W1}}{\text{T2} - \text{T1}}, \tag{3}$$

$$NAR \left( \mathrm{mg\,cm^{-2}\,day^{-1}} \right) = \frac{(W2 - W1)(Ln\,LA2 - Ln\,LA1)}{(LA2 - LA1)(T2 - T1)}. \tag{4}$$

$$LAD\,(\mathrm{day}) = \frac{(LAI1 + LAI2)(T2 - T1)}{2}, \tag{5}$$

$$LAR \left( \mathrm{cm^2\,g^{-1}} \right) = \frac{(LA2 - LA1)(Ln\,W2 - Ln\,W1)}{(Ln\,LA2 - Ln\,LA1)(W2 - W1)}, \tag{6}$$

$$CGR \left( \mathrm{mg\,cm^{-2}\,day^{-1}} \right) = LAI \times NAR. \tag{7}$$

where W1 and W2 are the PDW, while LA1 and LA2 are the leaf areas of plants in the first sample (T1) and second sample (T2), respectively.

### 2.3.2. Physiological Attributes

To record leaf relative water content (RWC), leaf samples from each subplot were randomly selected and their fresh weight (FW) was immediately recorded. Thereafter, the leaf samples were floated in distilled water under low light conditions for 24 h and then dried at 80 °C until a constant weight was reached, in order to record their turgid weights (TW) and dry weights (DW), respectively. The values of these three parameters were applied in the following formula to record the percentage of RWC:

$$RWC\,(\%) = (FW - DW)/(TW - DW) \times 100$$

The content levels of chlorophyll pigments, namely chlorophyll-a (Chl-a) and chlorophyll-b (Chl-b), were measured with spectrophotometry after fragments of fresh leaves (0.5 g) were extracted in 10 mL ethanol (95%). After complete extraction, the absorbance of the extracts was read using a spectrophotometer (UV-2550, Shimadzu, Tokyo, Japan) at 663 nm and 645 nm. Finally, the concentrations of the three parameters of chlorophyll pigments were calculated according to the method described previously by Lichtenthaler [39].

### 2.3.3. Yield and Crop Water Productivity

At maturity, five rows of wheat from the middle of each subplot were harvested by hand, air dried, and threshed to collect their grains. Then the grains were cleaned and weighed to calculate grain yield (GY). After the values for GY were converted to kg ha$^{-1}$, the crop water productivity (CWP) was calculated by dividing the GY by the amount of growing season irrigation water.

### 2.3.4. Stress Tolerance Indices

The different stress tolerance indices (STIs) were calculated for each IMPP based on the values of plant dry weight at 80 days after sowing (STIs-PDW-80), plant dry weight at 100 days after sowing (STIs-PDW-100), and grain yield (STIs-GY) under 1.00 ET and 0.50 ET treatments. The full name, abbreviation, and formula for each STI are presented in Table 1.

**Table 1.** The full names, abbreviations (Abb.), and formulas for the different stress tolerance indices (STIs).

| Full Name | Abb. | Formula |
| --- | --- | --- |
| Yield index | YI | S/Ś |
| Stress tolerance index | STI | (NS × S)/(NŚ)$^2$ |
| Stress sensitive index | SSI | 1 − (S/NS)/1 − (Ś/NŚ) |
| Geometric mean productivity index | GMP | $\sqrt{NS \times S}$ |
| Mean relative performance index | MRP | (S/Ś) + (NS/NŚ) |
| Relative efficiency index | REI | (S/Ś) × (NS/NŚ) |

S and NS are the values of the trait, for each treatment, as evaluated under full (1.00 ET) and limited (0.50 ET) irrigation conditions, respectively. Ś and NŚ are the mean values of all treatments evaluated under limited and full irrigation conditions, respectively.

### 2.4. Statistical Analysis

Before statistical analysis, the data values were subjected to homogeneity of variances and normality distribution tests using Bartlett's chi-square and the Shapiro–Wilk tests, respectively. Because a uniform error variance was detected for different parameters in the two growing seasons, a combined analysis of variance (ANOVA) was applied for the split-split plot design across two seasons. Seasons and replications were considered random effects, while irrigation and IMPPs were considered fixed effects. The mean values among the irrigation regimes and eight IMPPs treatments, as well as their interaction, were compared by the Duncan method at the 0.05 probability level. Regression relationship analyses were performed between different growth indicators as independent variables, and also using GY and WP as dependent variables, in order to identify the optimal values of the growth indicators that tend to maximize GY and WP under different IMPP practices. Associations between different IMPPs and studied parameters were determined through heatmap clustering analysis to identify the most efficient IMPP practices under each irrigation condition and across the two conditions. All figures were drawn using the Sigma Plot 14.0 software program (Systat software, Inc., Chicago, IL, USA).

## 3. Results

### 3.1. Vegetative Growth Parameters

The different growth parameters, i.e., PH, TN, GLN, GLA, GLDW, and PDW, as measured at 80 and 100 days after sowing (DAS), were significantly ($p < 0.001$) affected at a high level by the irrigation regime (IR), IMPPs, and their interaction (Table 2). The main effect of season (S) was significant ($p \leq 0.05$) for only GLN and GLA at 80 DAS. Two-way interactions between S and IR were significant for GLN, GLA, and PDW at 80 DAS and GLA, GLDW, and PDW at 100 DAS. Two-way interactions between S and IMPPs were significant only for GLA at 80 DAS, and GLA and GLDW at 100 DAS. The triple-interaction effects of IMPPs, IR, and S were only found for GLA at 100 DAS (Table 2).

**Table 2.** Effects of seasons, irrigation regimes, and IMPPs on different vegetative growth parameters of wheat at 80 and 100 days after sowing over two growing seasons.

| Studied Factor | | 80 Days after Sowing | | | | | | 100 Days after Sowing | | | | | |
|---|---|---|---|---|---|---|---|---|---|---|---|---|---|
| | | PH | TN | GLN | GLA | GLDW | PDW | PH | TN | GLN | GLA | GLDW | PDW |
| | | | | | | Season (s) | | | | | | | |
| Season 1 | | 68.80 a | 3.65 a | 7.88 b | 119.67 a | 0.696 a | 6.01 a | 77.11 a | 4.45 a | 6.48 a | 82.28 a | 0.564 a | 10.01 a |
| Season 2 | | 69.92 a | 3.97 a | 8.23 a | 98.37 b | 0.653 a | 6.34 a | 79.04 a | 4.51 a | 6.15 a | 77.11 a | 0.548 a | 9.81 a |
| | | | | | | Irrigation (IR) | | | | | | | |
| 1.00 ET | | 77.89 a | 4.24 a | 9.80 a | 134.11 a | 0.834 a | 6.85 a | 85.21 a | 5.00 a | 7.94 a | 100.56 a | 0.674 a | 12.17 a |
| 0.50 ET | | 60.82 b | 3.39 b | 6.31 b | 83.93 b | 0.515 b | 5.50 b | 70.94 b | 3.96 b | 4.69 b | 58.83 b | 0.438 b | 7.65 b |
| | | | | | Integrations of mulching and planting patterns (IMPPs) | | | | | | | | |
| FNM | | 65.95 d | 3.23 d | 6.93 d | 92.71 d | 0.563 e | 5.69 c | 73.88 cd | 3.94 d | 5.01 d | 63.98 d | 0.460 d | 8.33 d |
| FPFM | | 74.86 a | 4.41 a | 9.14 a | 131.91 a | 0.808 a | 6.70 a | 81.73 ab | 4.68 ab | 7.49 ab | 90.47 a | 0.650 a | 11.58 a |
| FCRM | | 71.20 bc | 4.11 b | 8.54 bc | 116.29 b | 0.724 bc | 6.54 a | 82.99 a | 4.87 a | 7.58 a | 92.45 a | 0.638 a | 11.13 b |
| RBNM | | 65.46 d | 3.29 d | 7.06 d | 91.93 d | 0.583 e | 5.57 c | 74.09 cd | 4.20 cd | 5.15 d | 64.87 cd | 0.462 d | 8.18 d |
| RBPFM | | 72.55 ab | 4.32 ab | 9.03 ab | 125.83 a | 0.756 b | 6.61 a | 79.95 b | 4.66 ab | 7.10 b | 88.39 a | 0.608 b | 11.25 ab |
| RBCRM | | 70.41 bc | 4.25 ab | 8.26 c | 117.28 b | 0.698 cd | 6.60 a | 82.24 ab | 4.93 a | 7.24 b | 91.02 a | 0.608 b | 10.96 b |
| RFNM | | 65.67 d | 3.27 d | 7.21 d | 89.79 d | 0.592 e | 5.59 c | 73.19 d | 4.12 d | 4.91 d | 69.16 c | 0.474 d | 8.38 d |
| RFPFM | | 68.77 cd | 3.63 c | 8.27 c | 106.41 c | 0.673 d | 6.13 b | 76.55 c | 4.47 bc | 6.05 c | 77.21 b | 0.548 c | 9.49 c |
| ANOVA | df | | | | | | | | | | | | |
| Season (S) | 1 | 0.259 ns | 0.054 ns | 0.017 * | 0.005 ** | 0.096 ns | 0.186 ns | 0.322 ns | 0.069 ns | 0.192 ns | 0.094 ns | 0.303 ns | 0.214 ns |
| IR | 1 | <0.001 *** | <0.001 *** | <0.001 *** | <0.001 *** | <0.001 *** | <0.001 *** | <0.001 *** | <0.001 *** | <0.001 *** | <0.001 *** | <0.001 *** | <0.001 *** |
| IR × S | 1 | 0.035 ns | 0.877 ns | 0.048 * | <0.001 *** | 0.849 ns | <0.001 *** | 0.385 ns | 0.077 ns | 0.152 ns | 0.025 * | 0.039 * | <0.001 *** |
| IMPPs | 7 | <0.001 *** | <0.001 *** | <0.001 *** | <0.001 *** | <0.001 *** | <0.001 *** | <0.001 *** | 0.001 *** | <0.001 *** | <0.001 *** | <0.001 *** | <0.001 *** |
| IMPPs × S | 7 | 0.826 ns | 0.660 ns | 0.361 ns | 0.008 ** | 0.694 ns | 0.868 ns | <0.950 ns | 0.964 ns | 0.888 ns | <0.001 *** | 0.023 * | 0.973 ns |
| IMPPs × IR | 7 | <0.001 *** | <0.001 *** | <0.001 *** | <0.001 *** | <0.001 *** | <0.001 *** | <0.001 *** | 0.001 *** | <0.001 *** | <0.001 *** | <0.001 *** | <0.001 *** |
| IMPPs × IR × S | 7 | 0.909 ns | 0.716 ns | 0.540 ns | 0.180 ns | 0.902 ns | 0.995 ns | <0.961 ns | 0.695 ns | 0.207 ns | 0.032 * | 0.344 ns | 0.943 ns |

F, RB, and RF represent flat, raised-bed, and ridge–furrow planting patterns, respectively, whereas NM, PFM, and CRM represent no-mulch, plastic film mulch, and crop residues mulch, respectively. PH, TN, GLN, GLA, GLDW, and PDW indicate plant height, tiller number, green leaf number, green leaf area, green leaf dry-weight, and plant dry-weight, respectively. Means with same letters in the same column are not significantly different ($p > 0.05$). *** Significant at $p < 0.0001$; ** Significant at $p < 0.001$; * Significant at $p < 0.05$; ns: not significant.

Regardless of the IMPPs, the 0.50 ET treatment exhibited a significant reduction in all growth parameters at both growth stages, compared with the 1.00 ET treatment. The PH, TN, GLN, GLA, GLDW, and PDW in the 0.50 ET treatment were 21.9%, 20.0%, 35.6%,

37.4%, 38.3%, and 19.7% (at 80 DAS), and 16.7%, 20.8%, 41.0%, 41.5%, 35.0%, and 37.2% (at 100 DAS), lower than in the 1.00 ET treatment, respectively (Table 2). Additionally, the different growth parameters also showed substantial differences among IMPP treatments regardless of the IR treatments. In general, the planting patterns of F and RB mulched with both plastic film (FPFM and RBPFM) and crop residues (FCRM and RBCRM) always exhibited the highest values of all growth parameters at both growth stages, whereas the corresponding planting patterns without mulching (FNM and RBNM), as well as the RF without mulching (RFNM) showed the lowest values of all growth parameters at both growth stages. Compared with non-mulched treatments, the RF mulched with plastic film (RFPFM) had a relatively small effect on the increase of growth parameters (Table 2). Compared with the different planting patterns without mulching (FNM, RBNM, and RFNM), the different growth parameters in planting patterns mulched with plastic film (FPFM and RBPFM) and crop residues (FCRM and RBCRM) increased by 10.9–29.0% and 7.2–22.0% at 80 DAS and by 8.8–31.2% and 10.8–32.2% at 100 DAS, respectively, whereas in RFPFM they increased by only 4.5–14.5% at 80 DAS, and by 3.7–17.0% at 100 DAS (Table 2).

The effects of different IMPP treatments on different growth parameters were also dependent on the IR treatments, as shown in Figure 3. In general, the planting patterns of F and RB mulched with plastic film or crop residues exhibited the highest values of all growth parameters under both irrigation conditions. The planting patterns of RF with or without mulching (RFNM and RFPFM) resulted in a lower value for all growth parameters than did the planting patterns of F and RB without mulching under 1.00 ET, whereas the opposite was true under 0.50 ET. Under 0.50 ET, the values of all growth parameters in the RFPFM were comparable to those in the planting patterns of F and RB mulched with plastic film (FPFM and RBPFM) or crop residues (FCRM and RBCRM) (Figure 3). Compared with the planting patterns of F and RB without mulching (FNM and RBNFM), the corresponding planting patterns mulched with plastic film (FPFM and RBPFM) increased the plant growth parameters by 8.0–23.5% and 14.6–40.7% (at 80 DAS), and by 4.2–24.6% and 13.5–44.8% (at 100 DAS), under 1.00 ET and 0.50 ET, respectively, and the corresponding planting patterns mulched with crop residues (FCRM and RBCRM) increased the plant growth parameters by 4.8–21.5% and 10.3–29.0% (at 80 DAS), and by 6.8–28.4% and 14.9–37.0% (at 100 DAS), under 1.00 ET and 0.50 ET, respectively. Although the RFNM and RFPFM exhibited the lowest values of all growth parameters under 1.00 ET conditions, the RFNM and RFPFM increased the different growth parameters by 9.5–25.0% and 14.2–40.7% (at 80 DAS), and by 6.9–32.3% and 13.4–44.1% (at 100 DAS), under 1.00 ET and 0.50 ET conditions, respectively, when compared with the planting patterns of F and RB without mulching (Figure 3).

### 3.2. Crop Growth Indicators

Table 3 shows that the different crop growth indicators, i.e., AGR, RGR, NAR, LAD, and LAR, as well as LAI and CGR, at 80 DAS (LAI-1 and CGR-1) and at 100 DAS (LAI-2 and CGR-2) were significantly ($p < 0.01$ and $<0.001$) affected to a high degree by IR, IMPPs, and their interaction. The S had a significant main effect on all crop growth indicators except for NAR, LAI-2, and CGR-2. The IR × S interaction had significant effects on LAD, LAI-1, LAI-2, and CGR-1, while the IMPPs × S and IR × IMPPs × S interactions had significant effects on LAI-1, LAI-2, LAD, and LAR (Table 3).

Averaged across IMPPs, a deficit irrigation treatment (0.50 ET) resulted in a reduction of 15.1–61.1% in different growth indicators, as compared to normal irrigation (1.00 ET) (Table 3). Regarding the effects of different IMPP treatments, the planting patterns mulched with plastic film (FPFM and RBPFM) or crop residues (FCRM and RBCRM) had the best effects on all crop growth indicators, followed by RF mulched with plastic film (RFPFM) (Table 3). Compared with FNM and RBNM, the planting patterns mulched with plastic film (FPFM and RBPFM) or crop residues (FCRM and RBCRM) increased the different crop growth indicators by 9.8–45.1% and 6.7–45.4%, respectively, whereas the RFPFM increased the different crop growth indicators by 7.2–23.7%. The planting pattern RFNM slightly

improved the different growth indicators. This treatment increased the different growth indicators by 1.4–14.8%, as compared with FNM and RBNM (Table 3).

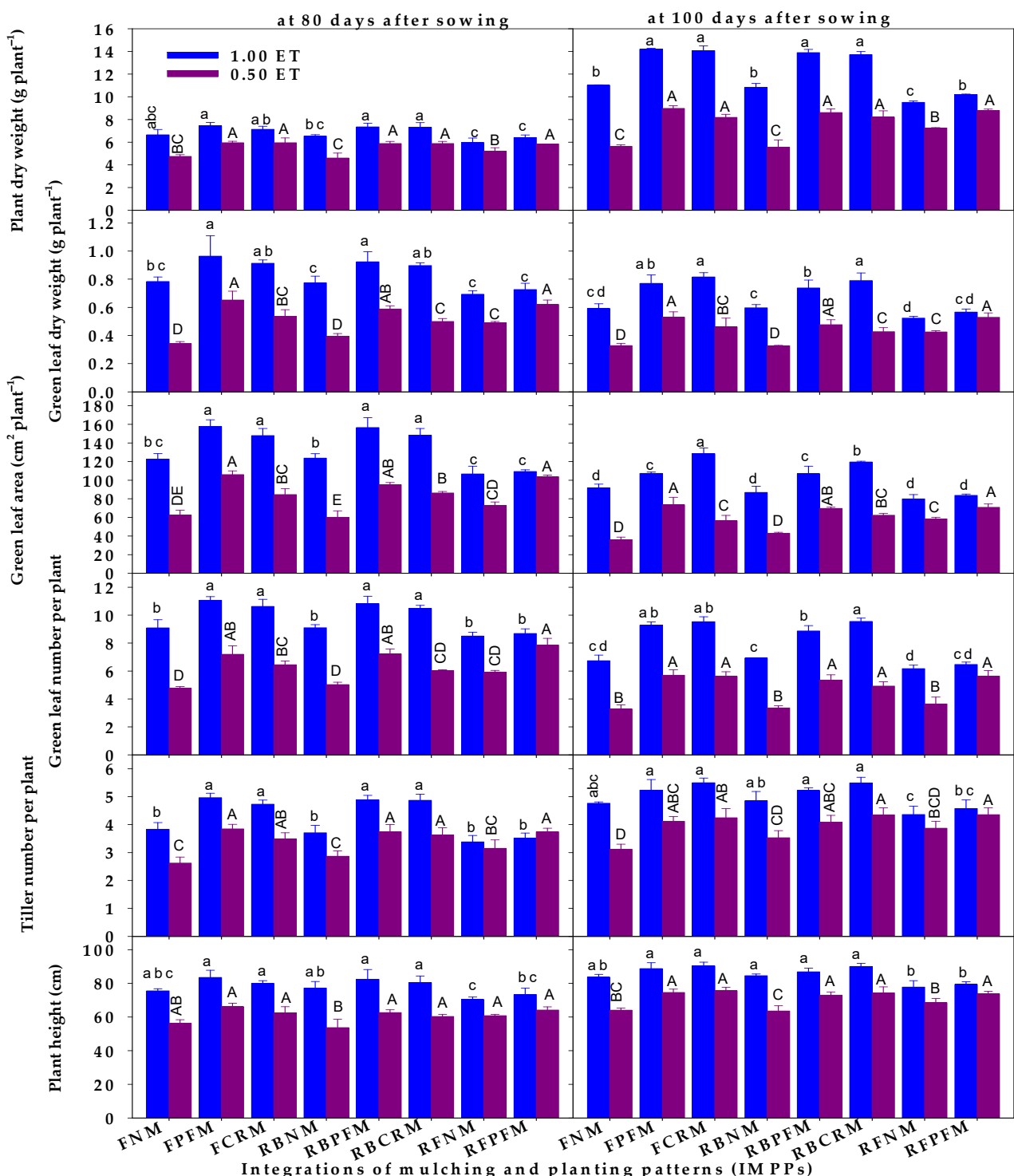

**Figure 3.** Effects of different integrations of mulching and planting patterns (IMPPs) on vegetative growth parameters, measured at 80 and 100 days after sowing under full (1.00 ET) and limited (0.50 ET) irrigation conditions. F, RB, and RF represent flat, raised-bed, and ridge–furrow planting patterns, respectively, whereas NM, PFM, and CRM represent no-mulch, plastic film mulch, and crop residues mulch, respectively. Lower-case and upper-case letters indicate significant differences among the eight IMPPs (Duncan's multiple range test, $p < 0.05$) under 1.00 ET and 0.50 ET treatments, respectively. Vertical bars represent the standard deviations of the means (n = 3).

**Table 3.** Effects of seasons, irrigation regimes, and IMPPs on different growth indicators, physiological attributes, grain yield (GY), and water productivity (WP) over two growing seasons.

| | | LAI-1 | LAI-2 | AGR | RGR | NAR | LAD | LAR | CGR-1 | CGR-2 | RWC | Chla | Chlb | Chlt | GY | WP |
|---|---|---|---|---|---|---|---|---|---|---|---|---|---|---|---|---|
| | | | | | | | Season (s) | | | | | | | | | |
| | Season 1 | 3.42 a | 2.35 a | 0.200 a | 0.024 a | 1.906 a | 57.70 a | 12.49 a | 6.80 a | 4.78 a | 78.92 a | 1.65 a | 0.67 a | 2.32 a | 5410.4 a | 1.135 a |
| | Season 2 | 2.81 a | 2.20 a | 0.173 b | 0.020 b | 1.813 a | 50.14 b | 10.75 b | 5.65 b | 4.34 a | 77.16 a | 1.35 b | 0.59 a | 1.98 b | 5655.2 a | 1.209 a |
| | | | | | | | Irrigation (IR) | | | | | | | | | |
| | 1.00 ET | 3.83 a | 2.87 a | 0.266 a | 0.028 a | 2.254 a | 67.05 a | 12.57 a | 8.76 a | 6.57 a | 83.97 a | 1.77 a | 0.76 a | 2.53 a | 6895.0 a | 1.061 b |
| | 0.50 ET | 2.40 b | 1.68 b | 0.107 b | 0.016 b | 1.464 b | 40.79 b | 10.67 b | 3.68 b | 2.55 b | 72.11 b | 1.27 b | 0.50 b | 1.77 b | 4170.6 b | 1.283 a |
| | | | | | | Integrations of mulching with planting patterns (IMPPs) | | | | | | | | | | |
| | FNM | 2.65 d | 1.83 d | 0.132 d | 0.017 d | 1.475 d | 44.77 d | 10.78 e | 4.45 e | 3.16 c | 74.06 d | 1.20 e | 0.55 d | 1.76 e | 5181.7 c | 1.077 e |
| | FPFM | 3.77 a | 2.58 a | 0.244 a | 0.026 a | 2.145 a | 63.54 a | 12.28 a | 8.43 a | 5.71 a | 80.75 a | 1.69 b | 0.66 abc | 2.35 b | 5900.0 a | 1.281 a |
| | FCRM | 3.32 b | 2.64 a | 0.229 ab | 0.025 a | 2.071 ab | 59.64 b | 11.68 bcd | 7.30 bc | 5.90 a | 79.48 a | 1.82 a | 0.70 ab | 2.52 a | 5790.8 ab | 1.189 bc |
| | RBNM | 2.63 d | 1.85 cd | 0.131 d | 0.017 d | 1.501 d | 44.80 d | 11.16 de | 4.52 e | 3.08 c | 75.36 cd | 1.24 e | 0.56 d | 1.80 e | 5185.8 c | 1.074 e |
| | RBPFM | 3.60 a | 2.53 a | 0.231 ab | 0.025 a | 2.099 a | 61.20 ab | 12.04 ab | 7.90 ab | 5.51 a | 79.70 a | 1.56 c | 0.62 bcd | 2.18 c | 5670.8 ab | 1.221 ab |
| | RBCRM | 3.35 b | 2.60 a | 0.218 b | 0.024 ab | 1.991 ab | 59.51 b | 11.83 abc | 7.02 c | 5.53 a | 78.68 ab | 1.75 b | 0.71 a | 2.46 a | 5639.2 b | 1.156 cd |
| | RFNM | 2.57 d | 1.98 c | 0.139 d | 0.020 c | 1.746 c | 45.41 d | 11.39 cd | 4.53 e | 3.50 c | 76.96 bc | 1.33 d | 0.59 cd | 1.93 d | 5160.8 c | 1.108 de |
| | RFPFM | 3.04 c | 2.21 b | 0.168 c | 0.022 bc | 1.844 bc | 52.46 c | 11.82 abc | 5.63 d | 4.09 b | 79.32 a | 1.56 c | 0.65 abc | 2.22 c | 5733.3 ab | 1.271 a |
| ANOVA | | | | | | | | | | | | | | | | |
| S.O.V | df | | | | | | | | | | | | | | | |
| S | 1 | 0.005 ** | 0.094 ns | 0.009 ** | 0.23 * | 0.083 ns | 0.003 ** | 0.007 ** | 0.019 * | 0.058 ns | 0.063 ns | 0.004 ** | 0.075 ns | 0.012 * | 0.174 ns | 0.097 ns |
| IR | 1 | <0.001 *** | <0.001 *** | <0.001 *** | <0.001 *** | <0.001 *** | <0.001 *** | 0.001 *** | <0.001 *** | <0.001 *** | <0.001 *** | <0.001 *** | <0.001 *** | <0.00 *** | <0.001 *** | <0.001 *** |
| IR × S | 1 | <0.001 *** | 0.025 * | 0.063 ns | 0.252 ns | 0.079 ns | 0.036 * | 0.651 ns | 0.008 ** | 0.240 ns | 0.001 ** | 0.368 ns | 0.924 ns | 0.577 ns | 0.016 * | 0.003 ** |
| IMPPs | 7 | <0.001 *** | <0.001 *** | <0.001 *** | <0.001 *** | <0.001 *** | <0.001 *** | <0.001 *** | <0.001 *** | <0.001 *** | <0.001 *** | <0.001 *** | 0.002 ** | <0.001 *** | <0.001 *** | <0.001 *** |
| IMPPs × S | 7 | <0.008 ** | <0.001 *** | 0.866 ns | 0.882 ns | <0.623 ns | 0.001 *** | <0.001 *** | 0.367 ns | 0.944 ns | 0.645 ns | <0.001 *** | 0.624 ns | <0.001 *** | 0.557 ns | 0.362 ns |
| IMPPs × IR | 7 | <0.001 *** | <0.001 *** | <0.001 *** | <0.001 *** | <0.002 ** | <0.001 *** | <0.001 *** | <0.001 *** | <0.001 *** | <0.001 *** | <0.001 *** | <0.003 ** | <0.001 *** | <0.001*** | <0.001*** |
| IMPPs × IR × S | 7 | <0.180 ns | <0.032 * | 0.983 ns | 0.965 ns | <0.944 ns | <0.046 * | 0.001 *** | 0.902 ns | <0.975 ns | 0.909 ns | <0.001 *** | 0.907 ns | <0.001 *** | 0.245 ns | 0.181 ns |

F, RB, and RF represent flat, raised-bed, and ridge–furrow planting patterns, respectively, whereas NM, PFM, and CRM represent no-mulch, plastic film mulch, and crop residues mulch, respectively. LAI-1, LAI-2, AGR, RGR, NAR, LAD, LAR, CGR-1, CGR-2, RWC, Chl-a, Chl-b, and Chlt indicate leaf area index, absolute growth rate, relative growth rate, net assimilation rate, leaf area duration, leaf area ratio, crop growth rate, relative water content, chlorophyll-a, chlorophyll-b, and total chlorophyll content, respectively. Values 1 and 2 represent measurements at 80 and 100 days after sowing, respectively. Means with the same letters and in the same column are not significantly different ($p > 0.05$). *** Significant at $p < 0.0001$; ** Significant at $p < 0.001$; * Significant at $p < 0.05$; ns: not significant.

The responses of different crop growth indicators to the different IMPP treatments varied with the IR, as shown in Figure 4. Under 1.00 ET conditions, the planting patterns mulched with plastic film (FPFM and RBPFM) or crop residues (FCRM and RBCRM) always exhibited the highest values of all crop growth indicators, whereas the corresponding planting patterns without mulching (FNM and RBNM) or planting patterns of RF with or without mulching (RFNM and RFPFM) showed the lowest values for all parameters. The values of different growth indicators were significantly increased by 6.4–40.7% in the former four IMPP treatments (FPFM, RBPFM, FCRM, and RBCRM) compared with the latter four IMPPs treatments (FNM, RBNM, RFNM and RFPFM) (Figure 4). Under 0.50 ET conditions, the different planting patterns mulched with plastic film (FPFM, RBPFM, and RFPFM) or crop residues (FCRM and RBCRM) achieved the highest values of all crop growth indicators, followed by RFNM. The three planting patterns mulched with plastic film, the two planting patterns mulched with crop residues, and RFNM enhanced the different crop growth indicators by 18.2–70.9%, 5.2–63.0%, and 9.1–61.9%, respectively, when compared with the two planting patterns without mulching (RFNM and RFPFM) (Figure 4).

### 3.3. Physiological Attributes

The IR and IMPPs, as well as their interaction, had a significant effect on all physiological attributes, viz., RWC, Chl-a, Chl-b, and total chlorophyll content (Chlt) measured at 100 DAS (Table 3). The S had a significant main effect on Chl-a and Chlt, whereas their interaction with IR had a significant effect on only RWC, and their interaction with IMPPs had a significant effect on Chl-a and Chlt. Triple-interaction effects of IMPPs, IR, and S were found for Chl-a and Chlt (Table 3).

Compared with normal irrigation treatment (1.00 ET), the 0.50 ET treatment significantly reduced RWC, Chl-a, Chl-b, and Chlt by 14.1%, 28.2%, 33.8%, and 29.9%, respectively (Table 3). Regarding the effects of different IMPPs treatments, compared with the planting patterns of F and RB without mulching (FNM and RBNM), the corresponding planting patterns mulched with plastic film (FPFM and RBPFM) or crop residues (FCRM and RBCRM) had the best effects on all physiological attributes, followed by the planting patterns of RF mulched with plastic film (RFPFM). Additionally, the RF without mulching (RFNM) had a relatively small effect on the increase of physiological attributes, as compared with the non-mulched F and RB planting patterns (Table 3). Compared with the non-mulched F and RB planting patterns, the corresponding planting patterns mulched with plastic film and crop residues, as well as RFPFM and RFNM, increased the RWC by 6.9%, 5.5%, 5.8%, and 2.9%, Chl-a by 24.9%, 31.6%, 21.8%, and 8.4%, Chl-b by 12.9%, 20.9%, 15.0%, and 6.0%, and Chlt by 21.5%, 28.6%, 19.8%, and 7.7%, respectively (Table 3).

The effects of different IMPP treatments on different physiological attributes also depended on the IR treatments, as shown in Figure 5. There was no significant difference in the RWC among IMPPs under 1.00 ET conditions, whereas, under 0.50 ET conditions, the RWC in the two planting patterns mulched with plastic film (FPFM and RBPFM), the two planting patterns mulched with crop residues (FCRM and RBCRM), RFPFM, and RFNM were 12.0%, 9.1%, 12.3%, and 6.3% higher, respectively, than those in the non-mulched planting patterns (FNM and RBNM) (Figure 5). Under 1.00 ET conditions, the highest values for the different chlorophyll attributes (Chl-a, Chl-b, and Chlt) were observed with the two planting patterns mulched with crop residues (FCRM and RBCRM), followed by the two planting patterns mulched with plastic film (FPFM and RBPFM), whereas the values of different chlorophyll attributes for RFPFM and RFNM were statistically on par with the non-mulched planting patterns FNM and RBNM (Figure 5). Under 1.00 ET conditions, the planting patterns mulched with crop residues (FCRM and RBCRM) and planting patterns mulched with plastic film (FPFM and RBPFM) increased Chl-a by 29.6% and 16.0%, Chl-b by 25.8% and 7.2%, and Chlt by 28.5% and 13.5%, respectively, when compared with the non-mulched planting patterns FNM and RBNM (Figure 5). Under 0.50 ET conditions, the highest values for the different chlorophyll parameters were observed with the RFPFM and the two planting patterns mulched with plastic film (FPFM and RBPFM), followed by the two planting patterns mulched

with crop residues (FCRM and RBCRM) (Figure 5). Compared with the two non-mulched treatments (FNM and RBNM), the data indicate that the RFPFM, the two treatments mulched with plastic film, and the two treatments mulched with crop residues increased Chl-a by 38.2%, 36.0%, and 34.6%, Chl-b by 26.6%, 20.6%, and 11.9%, and Chlt by 34.9%, 31.8%, and 28.7%, respectively. Although the RFNM was not mulched, the values of Chl-a, Chl-b, and Chlt in this planting pattern were 20.2%, 11.8%, and 17.7 higher, respectively, than those in the two non-mulched treatments (FNM and RBNM) (Figure 5).

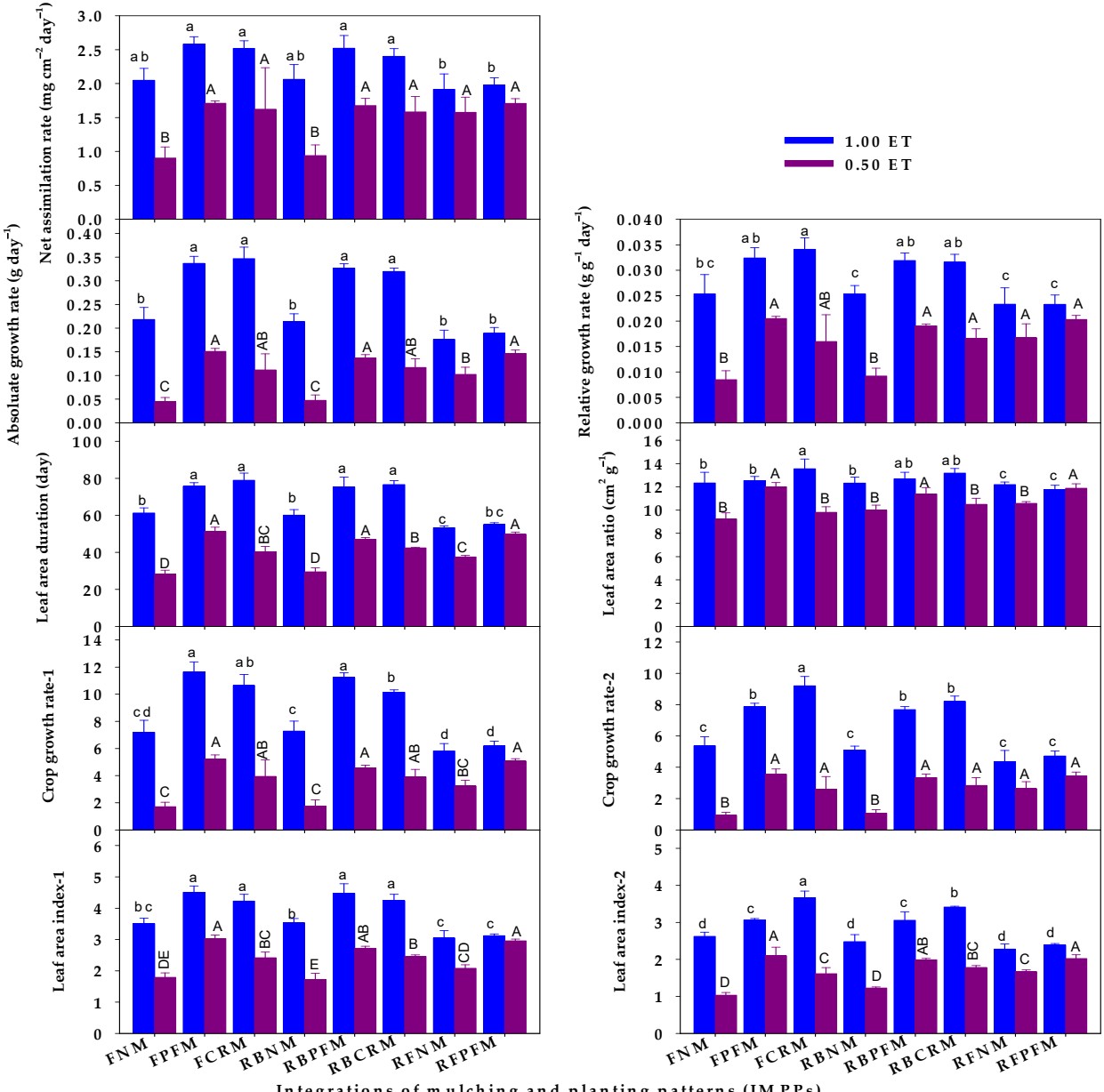

**Figure 4.** Effects of different integrations of mulching and planting patterns (IMPPS) on crop growth indicators under full (1.00 ET) and limited (0.50 ET) irrigation conditions. F, RB, and RF represent flat, raised-bed, and ridge–furrow planting patterns, respectively, whereas NM, PFM, and CRM represent no-mulch, plastic film mulch, and crop residues mulch, respectively. Values 1 and 2 represent measurements at 80 and 100 days after sowing, respectively. Lower-case and upper-case letters indicate significant differences among the eight IMPPs (Duncan's multiple range test, $p < 0.05$) under 1.00 ET and 0.50 ET treatments, respectively. Vertical bars represent the standard deviations of the means (n = 3).

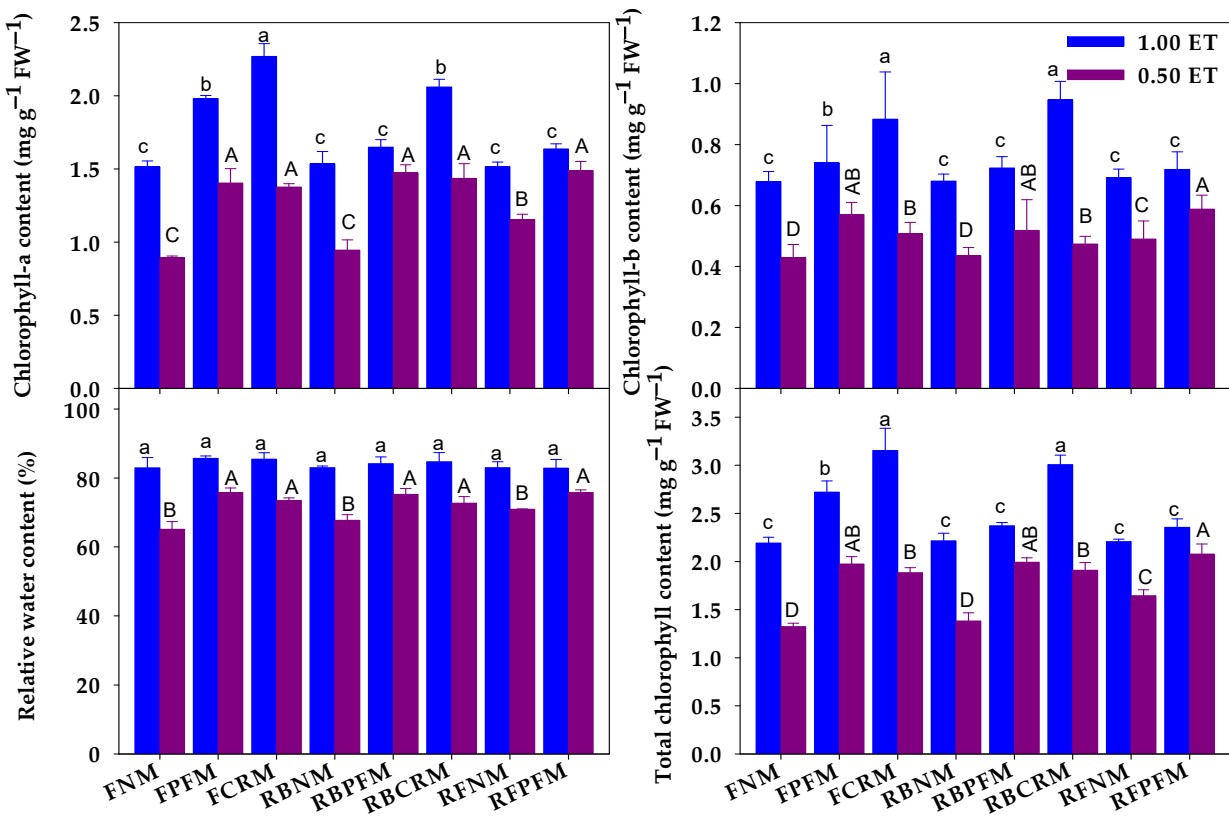

**Figure 5.** Effects of different integrations of mulching and planting patterns (IMPPs) on physiological attributes, measured at 100 days after sowing under full (1.00 ET) and limited (0.50 ET) irrigation conditions. F, RB, and RF represent flat, raised-bed, and ridge–furrow planting patterns, respectively, whereas NM, PFM, and CRM represent no-mulch, plastic film mulch, and crop residues mulch, respectively. Lower-case and upper-case letters indicate significant differences among the eight IMPPs (Duncan's multiple range test, $p < 0.05$) under 1.00 ET and 0.50 ET treatments, respectively. Vertical bars represent the standard deviations of the means (n = 3).

### 3.4. Grain Yield and Water Productivity

As shown in Table 3, the IR, the IMPPs, and the interaction of IMPPs × IR and IR × S had a significant effect on GY and WP, while there was no significant effect of S, two-way interaction of IMPPs × S, or three-way interaction of IMPPs × IR × S on both parameters.

The GY was superior under 1.00 ET as compared to 0.50 ET conditions; the opposite was true for WP (Table 3). The 1.00 ET treatment increased GY by 39.5%, as compared with the 0.50 ET treatment, while the later treatment increased WP by 17.3% as compared with the former treatment (Table 3). Regarding the IMPPs effects, the two planting patterns mulched with plastic film (FPFM and RBPFM), the two planting patterns mulched with crop residues (FCRM and RBCRM), and RFPFM exhibited the highest values of GY, while the values of GY for RFNM were statistically on par with the non-mulched planting patterns FNM and RBNM. The two planting patterns mulched with plastic film, two planting patterns mulched with crop residues, and RFPFM increased GY by 10.5%, 9.4%, and 9.7%, respectively, when compared with the non-mulched planting patterns FNM, RBNM, and RFNM (Table 3). The highest values for WP were observed with the RFPFM and the two planting patterns mulched with plastic film, followed by the two planting patterns mulched with crop residues. The values of WP were significantly increased by 14.5%, 13.2%, and 7.4% in these three planting patterns, respectively, when compared with the non-mulched planting patterns FNM, RBNM, and RFNM (Table 3).

The response of GY and WP to the different IMPPs also varied under each IR treatment (Figure 6). Under 1.00 ET, the highest values of GY were achieved with the two planting patterns mulched with plastic film (FPFM and RBPFM), followed by the two planting patterns mulched with crop residues (FCRM and RBCRM). However, the highest values of WP were observed with the planting patterns F and RB, with or without mulching (Figure 6). Under 0.50 ET, the highest values of GY and WP were obtained with the RFPFM, followed by the two planting patterns mulched with plastic film, and these three treatments (RFPFM, FPFM and RBPFM) increased GY or WP by 28.5%, 25.7%, and 20.2%, respectively, when compared with the non-mulched planting patterns FNM and RBNM. The values of GY or WP with the two planting patterns mulched with crop residues (FCRM and RBCRM) and RFNM were statistically on par with the non-mulched planting patterns FNM and RBNM (Figure 6).

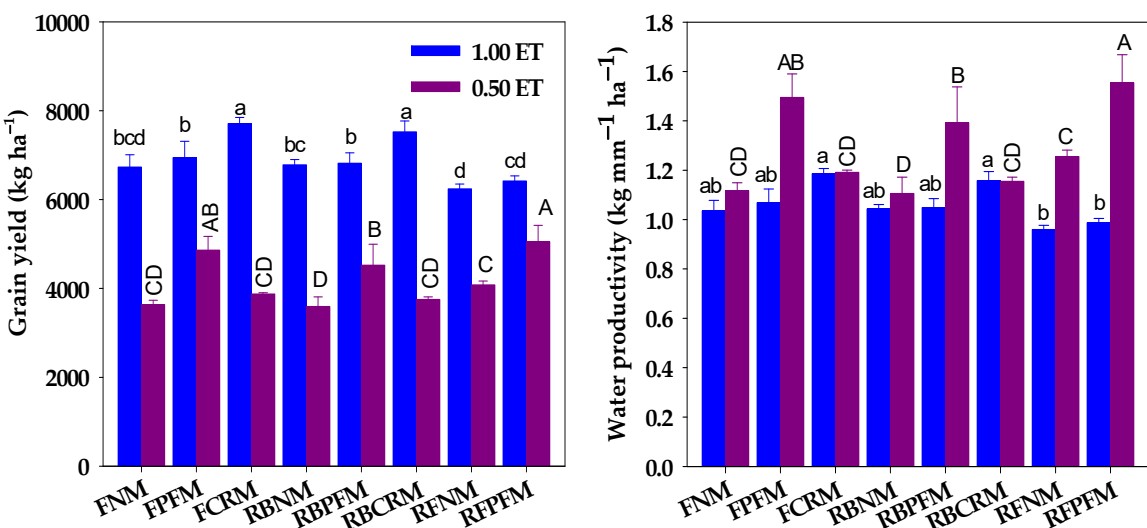

**Figure 6.** Effects of different integrations of mulching and planting patterns (IMPPs) on grain yield and water productivity under full (1.00 ET) and limited (0.50 ET) irrigation conditions. F, RB, and RF represent flat, raised-bed, and ridge–furrow planting patterns, respectively, whereas NM, PFM, and CRM represent no-mulch, plastic film mulch, and crop residues mulch, respectively. Lower-case and upper-case letters indicate significant differences among the eight IMPPs (Duncan's multiple range test, $p < 0.05$) under 1.00 ET and 0.50 ET treatments, respectively. Vertical bars represent the standard deviations of the means (n = 3).

*3.5. Stress Tolerance Indices*

To assess the ability of different IMPPs practices to enhance wheat performance and production, different stress tolerance indices based on PDW at 80 DAS (STIs-PDW-80), 100 DAS (STIs-PDW-100), and GY (STIs-GY) were calculated, and are presented in Figure 7. In general, different STIs revealed substantial differences between the different IMPP practices. The two planting patterns mulched with plastic film (FPFM and RBPFM) achieved the highest values for STIs based on PDW-80 and PDW-100, namely, yield index (YI), stress tolerance index (STI), mean relative performance (MRP), relative efficiency index (REI), and geometric mean productivity (GMP), followed by the two planting patterns mulched with crop residues (FCRM and RBCRM). However, the non-mulched planting patterns FNM and RBNM recorded the lowest values of the above-mentioned STIs (Figure 7). The planting pattern RFPFM exhibited the highest values of different STIs based on GY, and these values were comparable to those in two planting patterns mulched with plastic film, while their STIs based on PDW-80 and PDW-100 were lower than those in planting patterns mulched with plastic film or crop residues (Figure 7). The lowest values for stress sensitive index (SSI) were observed with RFPFM, while this index was high with the non-mulched

planting patterns, as well as with the two planting patterns mulched with crop residues, when calculated based on GY (Figure 7).

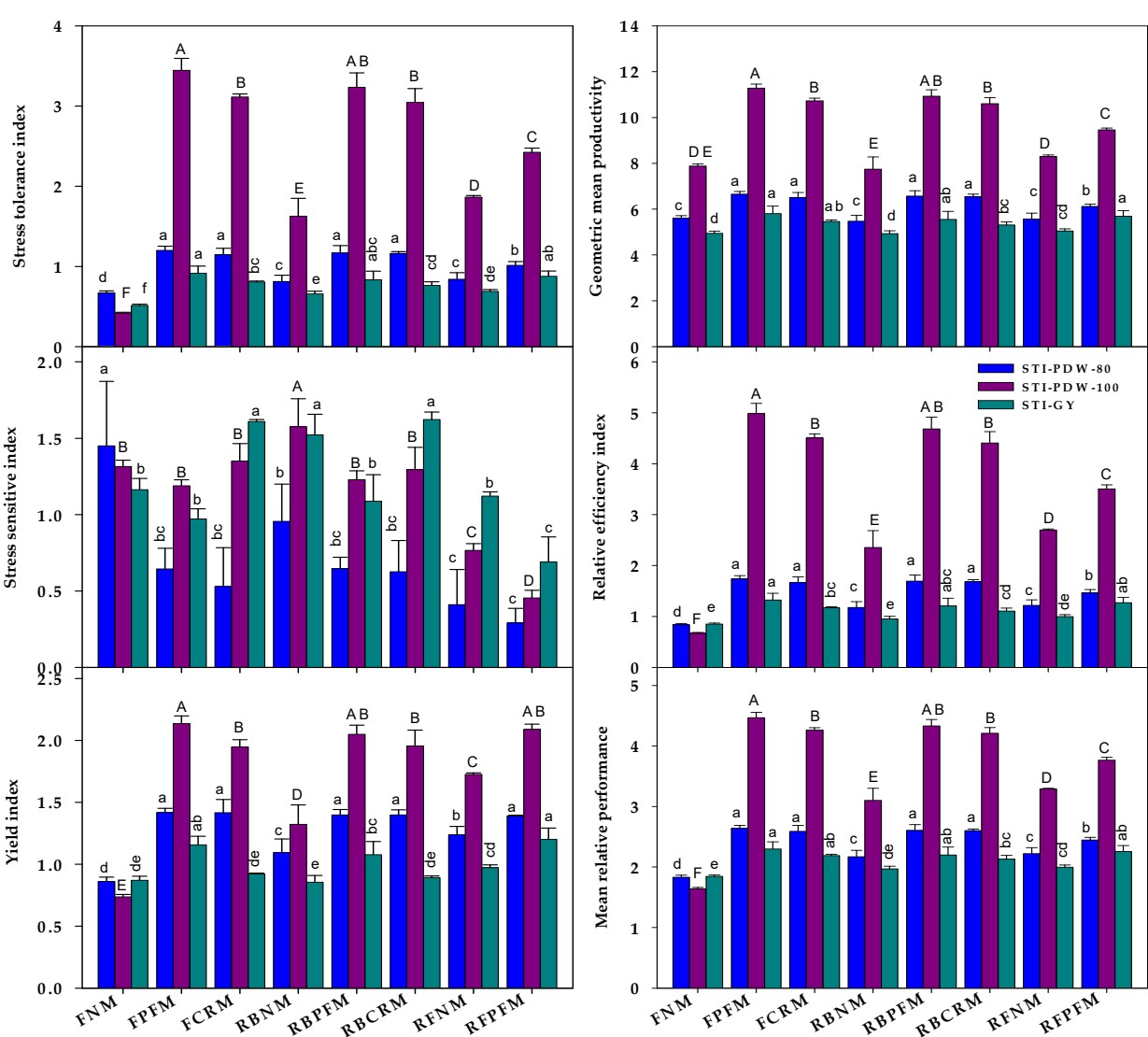

**Figure 7.** Comparison of different stress tolerance indices (STIs) among eight treatments with integrations of mulching and planting patterns (IMPPs). F, RB, and RF represent flat, raised-bed, and ridge–furrow planting patterns, respectively, whereas NM, PFM, and CRM represent no-mulch, plastic film mulch, and crop residues mulch, respectively. Lower-case and upper-case letters indicate significant differences among the eight IMPPs (Duncan's multiple range test, $p < 0.05$) for STIs based on plant dry-weight at 80 days after sowing (STIs-PDW-80), plant dry-weight at 100 days after sowing (STIs-PDW-100), and grain yield (STIs-GY), respectively. Vertical bars represent the standard deviations of the means (n = 3).

### 3.6. Relationship of Growth Indicators with Grain Yield and Water Productivity

The relationships of different growth indicators with GY and WP are presented in Figure 8. All growth indicators displayed a quadratic relationship with GY and WP, with the exception of RGR and NAR, which displayed a linear relationship with GY and WP (Figure 8). All growth indicators exhibited a strong relationship with GY ($R^2$ range 0.78 to 0.90). However, the WP showed a strong relationship with LAR ($R^2 = 0.81$) and CGR-2 ($R^2 = 0.70$), while it had a moderate relationship with other growth indicators ($R^2$ range 0.59 to 0.65) (Figure 8).

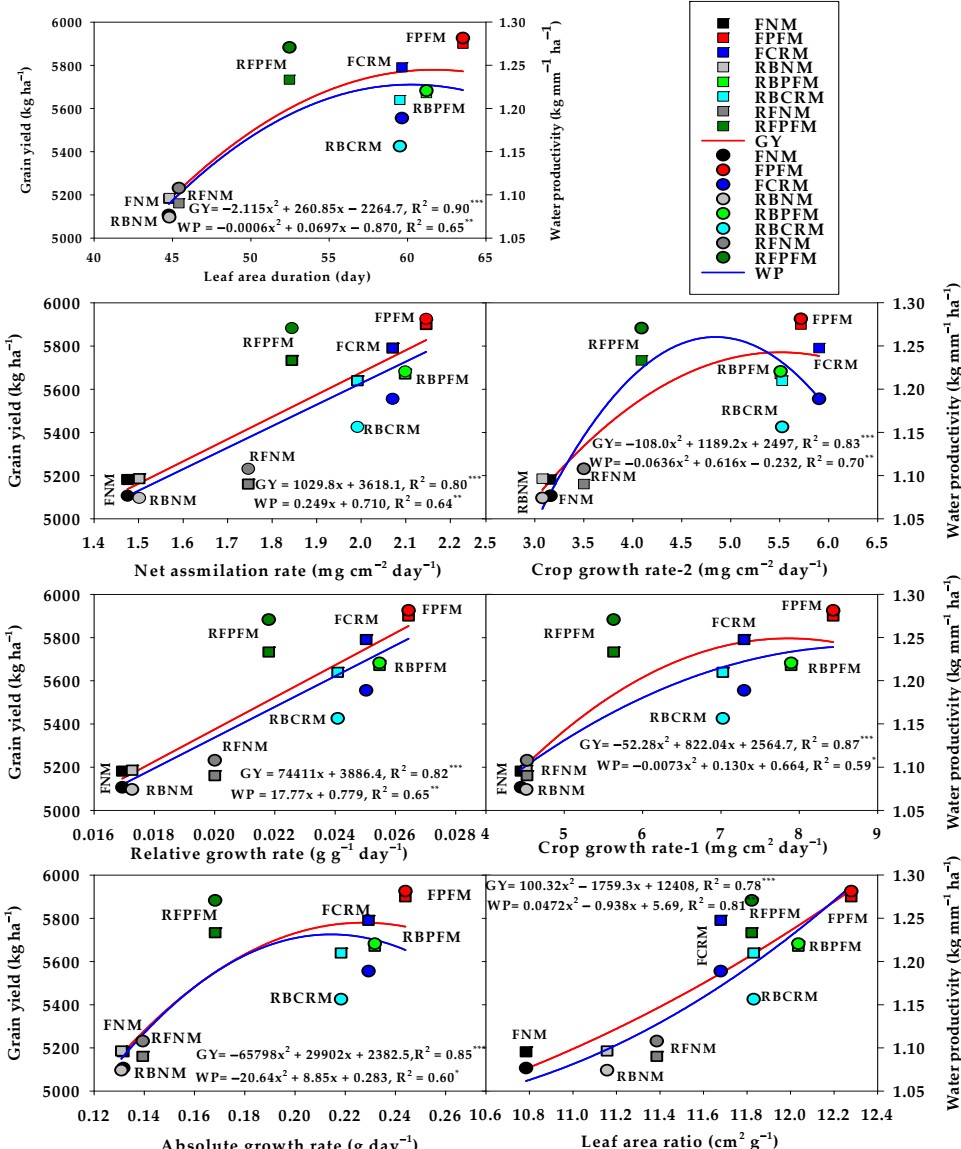

**Figure 8.** The fit regression models of different growth indicators with grain yield and water productivity for the eight treatments of integrated mulching and planting patterns (IMPPs). F, RB, and RF represent flat, raised-bed, and ridge–furrow planting patterns, respectively, whereas NM, PFM, and CRM represent no-mulch, plastic film mulch, and crop residues mulch, respectively. *** Significant at $p < 0.0001$; ** Significant at $p < 0.001$; * Significant at $p < 0.05$; ns: not significant.

Based on relationship slopes of different growth indicators with GY and WP, the optimum AGR, RGR, NAR, LAD, LAR, CGR-1, and CGR-2 values for maximizing GY were 0.23 g day$^{-1}$, 0.052 g g$^{-1}$ day$^{-1}$, 3.51 mg cm$^{-2}$ day$^{-1}$, 61.68 days, 8.77 cm$^{-2}$ g$^{-1}$, 7.86 mg cm$^{-2}$ day$^{-1}$, and 5.51 mg cm$^{-2}$ day$^{-1}$, respectively, and for maximizing WP were 0.21 g day$^{-1}$, 0.044 g g$^{-1}$ day$^{-1}$, 2.85 mg cm$^{-2}$ day$^{-1}$, 58.08 days, 9.94 cm$^{-2}$ g$^{-1}$, 8.91 mg cm$^{-2}$ day$^{-1}$, and 4.84 mg cm$^{-2}$ day$^{-1}$, respectively (Figure 8).

### 3.7. Selection of the Optimal IMPPs through Their Association with Studied Parameters

Heatmap dendrogram clustering was applied to select the best IMPPs under each irrigation condition, based on growth indicators, their associations with different vegetative growth parameters measured at 80 and 100 DAS, physiological attributes, GY, and WP (Figure 9A,B). The heatmap analysis based on all parameters divided the IMPPs into three main groups; each group was divided by irrigation condition (Figure 9A,B). Under 1.00 ET conditions, the planting patterns mulched with crop residues (FCRM and RBCRM) were clustered into one

group and showed the highest values for most parameters (red and yellow colors), followed by planting patterns mulched with plastic film (FPFM and RBPFM), which, clustered in one group, displayed the highest values for many parameters. Additionally, the three planting patterns without mulch (FNM, RBNM, and RFNM) and RF mulched with plastic film (RFPFM) were clustered together into one group and displayed the lowest values for all parameters (depicted in blue) (Figure 9A). However, under 0.50 ET conditions, the three planting patterns mulched with plastic film (FPFM, RBPFM, and RFPFM), clustered together into one group, displayed the highest values for all parameters (depicted in red and yellow colors), followed by planting patterns mulched with crop residues (FCRM and RBCRM) and RF without mulch (RFNM), which, clustered in one group, displayed medium values for all parameters. The two planting patterns without mulch (FNM and RBNM) were clustered together into one group and displayed the lowest values for all parameters (depicted in blue) (Figure 9B).

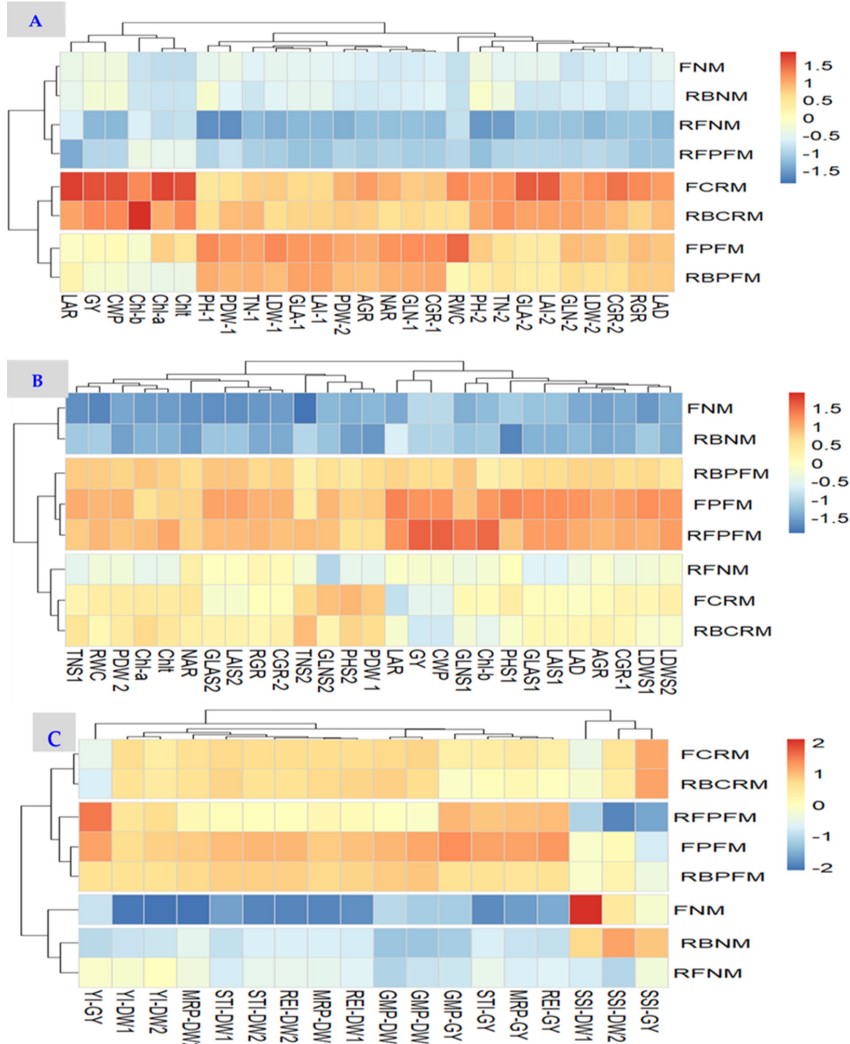

**Figure 9.** Heatmap and hierarchical clustering dividing the eight treatments of integrated mulching and planting patterns (IMPPs) into different clusters based on the assessed vegetative growth parameters of wheat at 80 (number 1) and 100 (number 2) days after sowing, growth indicators, physiological attributes, grain yield (GY), and water productivity (WP), under full (1.00 ET); (**A**) and limited (0.50 ET); (**B**) irrigation conditions, and based on different stress tolerance indices (STIs), irrespective of irrigation treatments. (**C**) F, RB, and RF represent flat, raised-bed (RB), and ridge–furrow planting patterns, respectively, whereas NM, PFM, and CRM represent no-mulch, plastic film mulch, and crop residues mulch, respectively. The full name of the abbreviations of all vegetative growth parameters are mentioned in the footnote of Table 1; growth indicators and physiological attributes are mentioned in the footnote of Table 2; and STIs are mentioned in Figure 7.

To select the best IMPPs in general, irrespective of IR treatments, the heatmap clustering was performed based on different STIs, which were calculated based on PDW measured at 80 and 100 DAS, as well as GY (Figure 9C). Similarly, the heatmap divided the IMPPs into four main groups. The three planting patterns mulched with plastic film (FPFM, RBPFM, and RFPFM) were clustered together into one group, and displayed the lowest values for SSI and the highest values for other STIs. The two planting patterns mulched with crop residues (FCRM and RBCRM) were clustered into one group and displayed the highest values for all STIs, including SSI. The planting pattern without mulch (FNM) was clustered alone into one group and displayed the highest values for SSI and the lowest values for other STIs. The other two planting patterns without mulch (RBNMM and RFNM) were separated from FNM and displayed the highest values for SSI and medium values for other STIs (Figure 9C).

## 4. Discussion

The results of this study showed that both IR and IMPPs practices had significant effects, individually as well as interactively, on growth performance, physiological attributes, production, and WP of wheat under arid conditions (Tables 2 and 3). These results reflect the finding that it is difficult to apply deficit irrigation to wheat plants without an accompanying negative impact on their growth performance and production under arid conditions. However, IMPP practices have played a vital role in overcoming these negative impacts of deficit irrigation. In general, regardless of the IR treatments, the results have shown that the two planting patterns mulched with plastic film (FPFM and RBPFM) seem to be superior in enhancing all studied parameters, followed by the two planting patterns mulched with crop residues (FCRM and RBCRM), when compared with the non-mulched planting patterns FNM and RBNM (Tables 2 and 3). A possible explanation for these findings is that plastic film mulching is usually able to reduce the amount of water lost through soil evaporation, which accounts for 40–60% of total crop water use [12,40,41], by preventing contact between the soil's surface and the atmospheric evaporation layer, as well as facilitating the movement of soil moisture from deeper soil layers to the active root zone for water absorption, thereby increasing the soil's water storage and prolonging the period of moisture availability for plants [8,16,25,36]. In addition, the plastic film mulching helps raise the topsoil temperature to a condition favorable for root growth during the early growth stages, thus increasing various root growth parameters such as root dry weight, root length density, and root surface area [42–44]. These improvements in root characteristics at early growth stages lead to the continuous availability of water and nutrients to plants. This ultimately leads to the rapid growth of early-stage wheat, which results in a prolonging of the functional period of green leaves, an increase in LAI and tiller numbers, increasing chlorophyll contents, promotion of the accumulation and transformation of photosynthetic products, and early formation for yield components, thereby enhancing the GY and WP [13,16,45–47]. This might explain why the two treatments mulched with plastic film (FPFM and RBPFM) in this study exhibited the highest values for different growth parameters, such as PH, TN, GLN, and LAI (Table 2), as well as growth indicators such as AGR, RGR, NAR, LAR, LAD, and CGR (Table 3). Similarly, Javaid et al. [16], Fan et al. [48] and Wu et al. [49] reported that warmer and wetter topsoil under plastic film mulch during the early growth stage promotes crop vegetative growth indicators, as indicated by greater PH, LAI, TN, CGR, biomass accumulation, GY, and water use efficiency.

The results of this study also indicate that the values of different parameters of the two planting patterns mulched with crop residues (FCRM and RBCRM) were occasionally comparable with the corresponding planting patterns mulched with plastic film (Tables 2 and 3). This may be because straw mulch, especially when chopped into small pieces, has a good ability to regulate the hydrothermal conditions of the soil by reducing soil evaporation and enhancing the soil's ability to withstand sudden oscillations in air temperature [24,26,50–52]. Furthermore, the decomposition of straw mulch during growing season successfully improves several physiochemical and biological properties of

soil, especially those associated with enhanced nutrient use efficiency, as well as improved soil water infiltration and retention [23,53,54]. The above-mentioned advantages of straw mulch contribute to a more favorable environment for crop growth, thus consequently promoting the different growth characteristics, physiological attributes, GY, and WP, as shown in the results of this study (Tables 2 and 3), which are consistent with the findings of numerous other studies [29,30,50,51].

An important finding of this study is that the effects of some IMPP practices on the growth, production, and WP of wheat depended on the irrigation rates. We found that, although the planting pattern RF mulched with plastic film (RFPFM failed to compete with FPFM and RBPFM in general (Tables 2 and 3) and under 1.00 ET treatment (Figures 3–6), it seemed to be superior in enhancing the growth characteristics, physiological attributes, GY, and WP of wheat under 0.50 ET treatment, as compared with FPFM, RBPFM, FCRM, and RBCRM (Figures 3–6). In addition, although the planting pattern RFNM did not use mulch, most of the studied parameters were relatively higher in this treatment than in non-mulched planting patterns FNM and RBNM under 0.50 ET treatment (Figures 3–6). A possible explanation for these findings is that the furrow where the wheat plants are grown has high efficiency in creating a deeper and more extensive root system and concentrating the limited amount of irrigation water applied within this root system. Furthermore, plastic film on ridges effectively reduces the amount of water lost from the soil through evaporation [15,16,31–34,36]. In addition, due to the nearness of the plant rows in the RF planting pattern, the furrow is always shaded by plants and ridges and thus receives less solar energy, which reduces the evaporation rate from the furrow and further improves the water availability in the soil layers [55,56]. However, on the other hand, the 50 cm wide ridge used in the RF planting pattern reduced the area allocated for wheat cultivation by 24% compared to the F planting pattern, while wheat was planted at the same rate. This increases the competition between plants under normal irrigation conditions as well as reduces the area allocated for harvesting by 24% [4,57]. All the above facts about the RF planting patterns with (RFPFM) or without mulching (RFNM) may explain why these two planting patterns were very effective under 0.50 ET conditions but failed to compete with any of the other IMPPs practices under 1.00 ET conditions (Figures 3–6).

The results of heatmap clustering, which provide a complete picture of the different IMPPs practices, based on their association with the various studied parameters, showed that crop residues mulching was more effective in enhancing most studied parameters than was plastic film mulching under 1.00 ET, and vice versa under 0.50 ET (Figure 9). These findings point out that crop residue mulching can be considered a useful practice when plants are grown under normal irrigation conditions; however, the benefits of this practice become limited when the amount of irrigation water is limited. A possible explanation for these findings is that the decompositions of crop residues by microbes during growing seasons, which become faster under high soil moisture content, can release a substantial amount of dissolved organic carbon and nitrogen, increase soil microbial activity by alleviating nutrient limitation of the microbes, release organic acids to the soil and reduce soil pH (which may increase the availability of many macro- and micro-nutrients and promote their absorption by plants), significantly increase flag leaf ion content (thus improving photosynthesis parameters, especially at the post-anthesis measurements), and delay the flag leaf senescence (which may support higher photosynthesis capacity during the grain filling period) [29,30,57–60]. However, due to the natural decomposition process for crop residues, the effectiveness of crop residue mulching at maintaining soil water content and reducing soil evaporation will decrease significantly over time [61,62]. These facts about crop residues mulching may explain why this practice was more effective than plastic film mulching under normal irrigation conditions, while less effective under limited irrigation conditions. Since the plastic film acts as a more effective insulation layer than crop residues and prevents the exchange of water between the soil and the air, it will be difficult for water in a root zone to turn into vapor and escape from the soil's surface. Therefore, plastic film mulching maintains soil water contents and extends the period of water availability for

plants [7,9,13,32,49]. This may explain why plastic film mulching treatments perform better in enhancing the growth, production, and WP of wheat under limited irrigation conditions, as shown in Figures 3–6. Similarly, a number of studies reported that plastic film mulching was more effective than straw mulching in counteracting water limitations in arid and semiarid regions [27,36,42,45].

The different growth indicators calculated in this study are considered to be a standard approach to explain the differences in growth potential and yield among various agronomic practices. These indicators reflect canopy development and the efficiency of solar radiation interception, thereby revealing the practices which can make a plant more or less productive, singly or in the population, under different agronomic practices. Therefore, several researchers have highlighted the relationship between GY and these growth indicators [63–65]. In this study, the quadratic curve was the best fit to explain the relationship of growth indicators with GY and WP (Figure 8). This result indicated that the ability of IMPP practices to improve GY and WP directly depends on the effectiveness of these practices for the enhancement of the growth performance of wheat plants during their early growth stages, particularly under limited irrigation conditions, through their ability to conserve more soil water content in the root zone, improve the nutrient availability for plants, and regulate topsoil temperature to a condition favorable for root growth during the early growth stages. The warmer and wetter topsoil under plastic film mulch during the early growth stage caused a rapid growth of wheat, as evidenced by high values of different vegetative growth parameters (Table 2). Such rapid growth causes more nutrient uptake during the early wheat-growth stage, which could result in high photosynthetic assimilation, maintain a relatively high LAI, delay leaf senescence, and increase LAD, thus subsequently resulting in high GY. Additionally, the crop residues not only reduced the amount of water lost through evaporation, but also helped regulate soil temperature during early growth stages, as well as improve the nutrient availability for plants during vegetative growth stages, which could result in enhancing several growth indicators, thus ultimately causing higher GY, particularly under normal irrigation conditions. Overall, these observations indicated that the IMPP practices that have the ability to enhance growth indicators during early growth stages also have the potential to enhance GY and WP during the reproductive growth stages.

## 5. Conclusions

This study was conducted to evaluate the impacts of different IMPP practices on the growth performance of a wheat crop in Saudi Arabia, as a typical arid country. The results of this study showed that the effects of mulching practices on vegetative growth at different growth stages, growth indicators, and physiological attributes of wheat varied with different planting patterns, mulching materials, and irrigation regimes. In general, the mulched treatments were more effective for enhancing the growth performance, GY, and WP of wheat than were non-mulched treatments. Additionally, the planting patterns mulched with crop residues were more effective in improving all studied parameters under normal irrigation conditions, while those mulched with the plastic film were helpful in enhancing the same parameters under deficit irrigation conditions. The planting pattern RF, with or without mulching, failed to compete with other IMPP practices under 1.00 ET treatment, while it seemed to be superior in enhancing the different studied parameters under deficit irrigation conditions. Collectively, we conclude that using PPs mulched with CRM is the recommended practice for achieving good performance and production of wheat under adequate irrigation, whereas using PPS mulched with PFM is recommended as a viable management option for sustainable production of wheat and improvement of WP under limited irrigation in arid countries.

**Author Contributions:** Conceptualization, S.E.-H., B.A., N.M. and Y.R.; methodology, S.E.-H., B.A. and N.M.; software, S.E.-H., B.A. and Y.R.; validation, S.E.-H., N.M. and B.A.; formal analysis, S.E.-H., N.M., Y.R. and B.A.; investigation, S.E.-H. and B.A.; resources, S.E.-H., B.A. and N.M.; data curation, S.E.-H., N.M., B.A. and Y.R.; writing—original draft preparation, S.E.-H.; writing—review and editing, S.E.-H.; visualization, S.E.-H.; supervision, S.E.-H.; project administration, S.E.-H.; funding acquisition, S.E.-H. All authors have read and agreed to the published version of the manuscript.

**Funding:** This research was funded by Deputyship for Research and Innovation, "Ministry of Education" in Saudi Arabia, research number (IFKSUOR3-106-1).

**Data Availability Statement:** All data are presented within the article.

**Acknowledgments:** The authors extend their appreciation to the Deputyship for Research and Innovation, "Ministry of Education" in Saudi Arabia for funding this research (IFKSUOR3-106-1).

**Conflicts of Interest:** The authors declare no conflict of interest.

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
