# Peer review of "Improving Morpho-Physiological Indicators, Yield, and Water Productivity of Wheat through an Optimal Combination of Mulching and Planting Patterns in Arid Farming Systems"

_agronomy, doi:10.3390/agronomy13061660_

Round 1

Reviewer 1 Report

Dear authors,

I must appreciate your thorough investigation and presentation of the research. However, I have noted the following limitations which can be considered.

Title: Title is appropriate.

Abstract: It can be reduced.

Keywords: Reduce keywords into 5 or 6

Introduction: Introduction is narrated nicely with the novelty of the study.

Materials & Methods:

In Experimental Design and Treatments segment (line 175), the design mentioned is RCBD (Line 176); however, the study was carried out in a split plot design and analysed statistically (as mentioned in line 289). Please talk to a statistician and correct it.

Check typos: Line 180

The plot size mentioned is 2m x 4m. It will be better to write 4m x 2m. Is it not too small?

Please mention the width of the plot bund.

Results

Table 1: Corrections suggested.

Subheading: 3.2 Crop growth indicators: There is a scope for briefing the content under the subheading.

Discussion: Line 780 to 790 and 843 to 845 can be briefed as it has been narrated earlier in the results section.

In general, the content narrated in the results section, should not be unnecessarily repeated (for discussion section).

References: Arrange the references as per the journal's guidelines.

All the best.

The English language is good; however, some typos and punctuation are to be thoroughly checked at the time of the revision.

Author Response

Reviewer #1

Dear authors,

I must appreciate your thorough investigation and presentation of the research. However, I have noted the following limitations which can be considered.

Response: We greatly appreciate your critical observations as well as your constructive and helpful comments. We hope that we could address your questions/comments by the explanations and revisions made in the manuscript. We believe that the manuscript is substantially improved after making the suggested revisions.

1- Title is appropriate.

Response: The title has been retained without modification.

2- Abstract: It can be reduced.

Response: Thank you very much for your suggestion. The abstract has been reduced.

3- Keywords: Reduce keywords into 5 or 6.

Response: Thank you very much for your suggestion. The keywords have been reduced into 6.

4- Introduction: Introduction is narrated nicely with the novelty of the study.

Response: Many thanks.

5- Materials & Methods: In Experimental Design and Treatments segment (line 175), the design mentioned is RCBD (Line 176); however, the study was carried out in a split plot design and analysed statistically (as mentioned in line 289). Please talk to a statistician and correct it.

Response: Thank you very much for your suggestion. You are right; the design of the experiment is a split-plot design with irrigation treatments and different IMPPs were allocated in the main plots and sub-plots, respectively. The design of the experiment has been rewritten as applied in the field.

6- Check typos: Line 180.

Response: A full-stop has been inserted.

7- The plot size mentioned is 2m x 4m. It will be better to write 4m x 2m. Is it not too small?

Please mention the width of the plot bund.

Response: Thank you very much for your suggestion. The plot size has been mentioned as 4m x 2m and the buffer zone between two adjacent subplots has been mentioned.

8- Here, you are mentioning Split Split plot design; however, in lines 176-177, you mentioned that RCBD. It's better you mention there also Split Split plot design

Response: Thank you very much for your comment. The design of the experiment has been rewritten accordingly.

9- Results. Table 1: Corrections suggested

Response: Thank you very much for your suggestion. Corrections have been made in Table 1 based on your suggestion.

10- Subheading: 3.2 Crop growth indicators: There is a scope for briefing the content under the subheading.

Response: Thank you very much for your suggestion. The content of Subheading 3.2 has been shortened.

11- Discussion: Line 780 to 790 and 843 to 845 can be briefed as it has been narrated earlier in the results section. In general, the content narrated in the results section, should not be unnecessarily repeated (for discussion section).

Response: Thank you very much for your suggestion. The lines 780 - 790 and 843 – 845 have been briefed.

12- Arrange the references as per the journal's guidelines.

Response: Thank you very much for your comment. The reference section has been arranged as per the journal's guidelines.

12- The English language is good; however, some typos and punctuation are to be thoroughly checked at the time of the revision.

Response: Thank you very much for editing the English language for our MS.

Reviewer 2 Report

 I have had the pleasure of reviewing your manuscript titled " Improving Morpho-Physiological Indicators, Yield, and Water Productivity of Wheat through an Optimal Combination of Mulching and Planting Patterns in Arid Farming Systems." I must commend your comprehensive and robust approach to this pertinent issue in the realm of agricultural science.

The issue of water productivity is indeed a pressing one, and your research not only highlights the problem but provides a tangible and innovative solution. Your study design, especially the combination of mulching and planting patterns, are appropriately rigorous, enabling a holistic understanding of the variables at play. Your focus on a wide array of measurements, including growth, productivity, physiological, and biochemical parameters, adds depth to the analysis. However, there are some recommended amendments are required as follow:

Point 1: The title needs to be amended to include “under water stress”

Point 2: The attention to detail in understanding the impacts of the combination of mulching and planting patterns on water productivity (WP), chlorophyll contents is highly commendable. However, SPAD value, and chlorophyll a fluorescence features (Fv/Fm, Fv/F0 and PI), plant water status (RWC, MSI, and EL) not only the RWC measured in your study, as well as the stomatal conductance are highly recommended to be measured under water stress scenarios.

Point 3: My second comment pertains to the largely descriptive nature of your results. While the data you present offer valuable insights, there could be additional analyses to deepen the interpretation and implications of your findings. Specifically, I recommend complementing your trait-by-trait analyses with correlation analysis and Principal Component Analysis (PCA).

Incorporating these statistical tools would not only strengthen your results but also provide a more nuanced understanding of the data. It would contribute to a richer and more comprehensive discussion of the interplay between mulching and planting patterns, water stress, and the physiological and biochemical responses of the wheat plants.

Point 4: While your manuscript clearly reflects a sound understanding of the existing literature, I would like to emphasize the importance of including more recent publications in your references. Scholarly conversations are continually evolving and including citations from the last three years (especially from 2022 and 2023) will ensure your work is positioned within the most current state of the field.

Point 5: Another aspect of your manuscript that could benefit from further clarification pertains to the error values depicted in your figures. While the inclusion of error bars is commendable as it provides a visual representation of variability or uncertainty, it is important to specify what these error values represent in the figure captions.

Error values can represent a range of statistical measures, including confidence intervals, standard errors, standard deviations, or other quantities. Each of these types of error bars conveys different information about your data. For example, standard deviation bars provide insight into the spread of the data, while standard error bars or confidence intervals offer a sense of the precision of the mean estimate.

Point 6: In addition to the points I've raised previously, it's crucial to address the need for error values in your data tables. Providing error values alongside the mean values in your tables is necessary to give an accurate representation of your data's variability and precision. The inclusion of error values, such as standard deviations, standard errors, or confidence intervals, helps to convey the spread and reliability of your data.

Point 7: I noticed that further information should be added in the M&M section to describe stress tolerance indices (stress tolerance index, stress sensitive index, Geometric mean production, relative efficiency index, and mean relative performance ) illustrated in section 3.5. and presented in Figure 7.

Point 8: In your manuscript, I noticed that further information could be beneficial in the captions of your tables and figures, particularly regarding the number of replicates used in your experiments.

Including information about the number of replicates (n) in your captions is important for several reasons. Firstly, it provides the reader with a clear understanding of the sample size, which can influence the interpretation of the data, especially when considering variability and statistical significance. Secondly, it provides context for the error bars (if present) and helps to validate the statistical analysis conducted. Lastly, it adds to the overall transparency and reproducibility of your research, both of which are critical aspects of scientific inquiry.

Point 9: Your manuscript would benefit from some minor revisions to ensure accuracy and consistency in your citations, references, figures, tables, and abbreviations. Here are the details:

a.       Cross-Reference Citations: Please verify that all the references in your bibliography have been cited within the body of the text and vice versa. Ensuring a match between the cited literature and the reference list will eliminate potential confusion for readers and maintain the integrity of your scholarly work.

b.       Author Name and Date Accuracy: Please double-check the spelling of the authors' names and the dates in your citations and references. These details should be correct and consistent throughout the manuscript to ensure proper attribution and to facilitate further reading by interested readers.

c.       Full Journal Titles: In the references section, kindly spell out all journal titles in full. Using abbreviations might lead to ambiguity or confusion, especially for lesser-known journals. Writing out full journal titles will ensure that your references are clear and easily accessible for readers.

d.       Figures and Tables Citations: Please ensure that all figures and tables are cited within the text and that these citations occur in consecutive order, corresponding to the appearance of figures and tables in the manuscript. This sequential citation will make it easier for readers to follow along and understand the relevance of each figure or table to your discussion.

e.       Abbreviations: Lastly, make sure to spell out all abbreviations in full the first time they appear in the text, followed by the abbreviation in parentheses. This practice will ensure clarity for all readers, especially those who might not be familiar with specific abbreviations.

Point 10: While your manuscript displays a strong command of the topic and presents compelling findings, I noticed some minor language issues and inconsistencies throughout the text. These could potentially hinder the clarity of your message and disrupt the reader's engagement with your work.

Author Response

Reviewer #2

Dear authors,

I have had the pleasure of reviewing your manuscript titled "Improving Morpho-Physiological Indicators, Yield, and Water Productivity of Wheat through an Optimal Combination of Mulching and Planting Patterns in Arid Farming Systems." I must commend your comprehensive and robust approach to this pertinent issue in the realm of agricultural science.

The issue of water productivity is indeed a pressing one, and your research not only highlights the problem but provides a tangible and innovative solution. Your study design, especially the combination of mulching and planting patterns, are appropriately rigorous, enabling a holistic understanding of the variables at play. Your focus on a wide array of measurements, including growth, productivity, physiological, and biochemical parameters, adds depth to the analysis. However, there are some recommended amendments are required as follow:

Response: We greatly appreciate your critical observations as well as your constructive and helpful comments. We hope that we could address your questions/comments by the explanations and revisions made in the manuscript. We believe that the manuscript is substantially improved after making the suggested revisions.

1- The title needs to be amended to include “under water stress”.

Response: Thank you very much for your suggestion. But, if we added “under water stress”, the title becomes long. In addition, based on the results of this study, the IMPPs treatments added benefit results under both adequate (1.00 ET) and limited (0.50 ET) irrigation conditions    

2- The attention to detail in understanding the impacts of the combination of mulching and planting patterns on water productivity (WP), chlorophyll contents is highly commendable. However, SPAD value, and chlorophyll a fluorescence features (Fv/Fm, Fv/F0 and PI), plant water status (RWC, MSI, and EL) not only the RWC measured in your study, as well as the stomatal conductance are highly recommended to be measured under water stress scenarios.

Response: Thank you very much for your comment. In this manuscript, the eight IMPPs treatments were evaluated from an agronomic point of view, and in a future study, these treatments could be evaluated from a physiological point of view under drought stress conditions.    

3- My second comment pertains to the largely descriptive nature of your results. While the data you present offer valuable insights, there could be additional analyses to deepen the interpretation and implications of your findings. Specifically, I recommend complementing your trait-by-trait analyses with correlation analysis and Principal Component Analysis (PCA). Incorporating these statistical tools would not only strengthen your results but also provide a more nuanced understanding of the data. It would contribute to a richer and more comprehensive discussion of the interplay between mulching and planting patterns, water stress, and the physiological and biochemical responses of the wheat plants.

Response: Thank you very much for your suggestion. As this study focused on the importance of IMPPs treatments in promoting wheat growth and production, we found that heatmap analysis was more appropriate than principal components analysis (PCA) to achieve this goal. Moreover, since the PCA analysis gave the same trend for the parameters as the heatmap analysis, only the heatmap analysis was presented in the manuscript. Importantly, because different growth indicators (AGR, RGR, NAR, LAD, LAR, and LAI and CGR) were computed from different growth parameters, it would naturally be that there were strong correlations between them. Because different growth indicators explain the effects of IMPPs on the growth potential of plants between the two growth stages, we found it best to make relationships between these growth indicators and GY and WP to show the importance of each of the eight IMPPs treatments.  

4- While your manuscript clearly reflects a sound understanding of the existing literature, I would like to emphasize the importance of including more recent publications in your references. Scholarly conversations are continually evolving and including citations from the last three years (especially from 2022 and 2023) will ensure your work is positioned within the most current state of the field.

Response: Thank you very much for your suggestion. All recent publications related to our study are already mentioned in the manuscript. Unfortunately, we did not find publications in 2023 related to our study to be cited in MS. All publications in 2022 related to this study were cited in MS.   

5- Another aspect of your manuscript that could benefit from further clarification pertains to the error values depicted in your figures. While the inclusion of error bars is commendable as it provides a visual representation of variability or uncertainty, it is important to specify what these error values represent in the figure captions. Error values can represent a range of statistical measures, including confidence intervals, standard errors, standard deviations, or other quantities. Each of these types of error bars conveys different information about your data. For example, standard deviation bars provide insight into the spread of the data, while standard error bars or confidence intervals offer a sense of the precision of the mean estimate.

Response: Thank you very much for your comment. The vertical bars represent the standard deviations of the means (n = 3). The vertical bars have been specified in the captions of Figures (3-7).

6- In addition to the points I've raised previously, it's crucial to address the need for error values in your data tables. Providing error values alongside the mean values in your tables is necessary to give an accurate representation of your data's variability and precision. The inclusion of error values, such as standard deviations, standard errors, or confidence intervals, helps to convey the spread and reliability of your data.

Response: Thank you very much for your comment. We have attempted to add the standard deviations after each mean, but unfortunately, the column width is not sufficient to include the means ± SD and the alphabet letters for significance. In addition, if we add ± SD, the tables become compact.

7- I noticed that further information should be added in the M&M section to describe stress tolerance indices (stress tolerance index, stress sensitive index, Geometric mean production, relative efficiency index, and mean relative performance ) illustrated in section 3.5. and presented in Figure 7.

Response: Thank you very much for your comment and reminder to add equations for stress tolerance indices (STIs). The subheading “2.3.4." has been added to the M&M section and related to Stress Tolerance Indices” has. The full name, abbreviation, and formula for each STI have been presented in Table 1.   

 8- In your manuscript, I noticed that further information could be beneficial in the captions of your tables and figures, particularly regarding the number of replicates used in your experiments.

Including information about the number of replicates (n) in your captions is important for several reasons. Firstly, it provides the reader with a clear understanding of the sample size, which can influence the interpretation of the data, especially when considering variability and statistical significance. Secondly, it provides context for the error bars (if present) and helps to validate the statistical analysis conducted. Lastly, it adds to the overall transparency and reproducibility of your research, both of which are critical aspects of scientific inquiry.

Response: Thank you very much for your comment and suggestion. The number of replications (n) has been mentioned in the captions of each Figure.

9- Your manuscript would benefit from some minor revisions to ensure accuracy and consistency in your citations, references, figures, tables, and abbreviations. Here are the details:.

  • Cross-Reference Citations: Please verify that all the references in your bibliography have been cited within the body of the text and vice versa. Ensuring a match between the cited literature and the reference list will eliminate potential confusion for readers and maintain the integrity of your scholarly work

Response: Thank you very much for your suggestion. Matching between the cited literature and the reference list has been carefully checked

  • Author Name and Date Accuracy: Please double-check the spelling of the authors' names and the dates in your citations and references. These details should be correct and consistent throughout the manuscript to ensure proper attribution and to facilitate further reading by interested readers.

Response: Done

  • Full Journal Titles: In the references section, kindly spell out all journal titles in full. Using abbreviations might lead to ambiguity or confusion, especially for lesser-known journals. Writing out full journal titles will ensure that your references are clear and easily accessible for readers.

Response: Thank you very much for your suggestion. The references section has been prepared according to the journal's guidelines. Based on the journal guidelines, the name of the journals should be written in abbreviation and not in the full name.

  • Figures and Tables Citations: Please ensure that all figures and tables are cited within the text and that these citations occur in consecutive order, corresponding to the appearance of figures and tables in the manuscript. This sequential citation will make it easier for readers to follow along and understand the relevance of each figure or table to your discussion.

Response: Done.

  • Abbreviations: Lastly, make sure to spell out all abbreviations in full the first time they appear in the text, followed by the abbreviation in parentheses. This practice will ensure clarity for all readers, especially those who might not be familiar with specific abbreviations.

Response: Done.

10- While your manuscript displays a strong command of the topic and presents compelling findings, I noticed some minor language issues and inconsistencies throughout the text. These could potentially hinder the clarity of your message and disrupt the reader's engagement with your work.

Response: Thank you very much for your valuable comment. The English language of MS has been carefully checked.   
